# Tracking matricellular protein SPARC in extracellular vesicles as a non-destructive method to evaluate lipid-based antifibrotic treatments

Cristina Zivko [1,2,3], Kathrin Fuhrmann[3], Gregor Fuhrmann [3,4✉] & Paola Luciani [1,2✉]

Uncovering the complex cellular mechanisms underlying hepatic fibrogenesis could expedite the development of effective treatments and noninvasive diagnosis for liver fibrosis. The biochemical complexity of extracellular vesicles (EVs) and their role in intercellular communication make them an attractive tool to look for biomarkers as potential alternative to liver biopsies. We developed a solid set of methods to isolate and characterize EVs from differently treated human hepatic stellate cell (HSC) line LX-2, and we investigated their biological effect onto naïve LX-2, proving that EVs do play an active role in fibrogenesis. We mined our proteomic data for EV-associated proteins whose expression correlated with HSC treatment, choosing the matricellular protein SPARC as proof-of-concept for the feasibility of fluorescence nanoparticle-tracking analysis to determine an EV-based HSCs' fibrogenic phenotype. We thus used EVs to directly evaluate the efficacy of treatment with S80, a polyenylphosphatidylcholines-rich lipid, finding that S80 reduces the relative presence of SPARC-positive EVs. Here we correlated the cellular response to lipid-based antifibrotic treatment to the relative presence of a candidate protein marker associated with the released EVs. Along with providing insights into polyenylphosphatidylcholines treatments, our findings pave the way for precise and less invasive diagnostic analyses of hepatic fibrogenesis.

[1] Institute of Pharmacy, Friedrich Schiller University of Jena, Jena, Germany. [2] Department of Chemistry, Biochemistry and Pharmaceutical Sciences, University of Bern, Bern, Switzerland. [3] Helmholtz Institute for Pharmaceutical Research Saarland, Department of Pharmacy, Saarland University, Saarbrücken, Germany. [4] Present address: Department of Biology, Friedrich-Alexander-University Erlangen, Erlangen, Germany. ✉email: gregor.fuhrmann@fau.de; paola.luciani@unibe.ch

The liver plays multifaceted roles in many physiologically critical functions: from protein syntheses to xenobiotic biotransformation, to immunological support[1,2]. Diseases compromising this essential organ directly lead to the death of 2 million people worldwide every year, i.e., 3.5% of all yearly deaths[3–5]. The global health burden of hepatic conditions has yet to be adequately met[6,7].

Under physiologically healthy conditions, hepatic stellate cells (HSCs) store vitamin A in cytoplasmic lipid droplets. Upon liver insults, however, they undergo transdifferentiation and become activated, losing their lipid droplets and starting to promote fibrogenesis[8–10]. In chronic conditions, the escalating deposition of excess, collagen-rich extracellular matrix leads to cirrhosis, and eventually to organ failure[11,12]. The current gold standard for the diagnosis of liver fibrosis is tissue biopsy, although alternative (albeit imprecise) methods are being explored, such as those based on ultrasonography methods, as well as those relying on clinical parameters[13–17]. The search for sensitive, precise, and noninvasive tools for the evaluation of liver fibrosis and its progression is an open field of investigation. This is particularly important because of liver fibrosis's mostly asymptomatic progression in its early and crucial stages[18,19].

Extracellular vesicles (EVs) are membranous nanosized vesicles mediating intercellular communication, secreted by virtually all cells[20–23]. They are increasingly being investigated for their potential as diagnostic tools given the rich differences that can arise in their biochemical composition as well as in their cargo[24–26]. Methods to work with EVs can be as varied as the research groups devoted to developing them[2], but the international community has been trying to push for a more rigorous standardization of protocols[27].

Here we established a robust series of methods for the isolation and thorough characterization of EVs originating from differently treated LX-2, an immortalized human HSC line retaining key features of transdifferentiated human HSCs[28], building upon an in vitro model for liver fibrosis we have optimized before[29]. We evaluated the effect exerted by EVs isolated from previously treated HSCs onto naïve cells. We investigated differences in the lipid composition of EVs and their cells of origin. Finally, we mined our EV-associated proteomic data for the development of treatment-discriminating tools. We established a convenient method to reliably detect exosomal markers CD81 and CD9[27], as well as rationally selected, cell status-discriminating proteins, using fluorescence nanoparticle-tracking analysis (f-NTA). We chose the secreted protein acidic and cysteine-rich (SPARC), supported by previous reports of its role in fibrogenic processes[30–32]. SPARC presence on EVs, as evidenced by our f-NTA method, could then be used to evaluate treatment response, most notably that of the polyenylphosphatidylcholines (PPCs)-rich lipid SPC, on which potential beneficial effects we have reported before[29]. Essential phospholipids (PPCs-enriched soybean extracts), have long been indicated as supportive therapy for liver diseases even though their mechanism of action is not well understood[33–35]. In this study, we could use biochemical information from EVs to assess the performance of antifibrotic PPCs, and also provide insights into their mode of action.

## Results

**EV-isolation, purification, and characterization.** In order to establish the optimal experimental setup (Fig. 1), EVs from untreated (DMEM-treated), quiescent-like (treated with retinol, ROL, and palmitic acid, PA), and perpetuated LX-2 cells (treated with transcriptional growth factor β1, TGF) were analyzed in terms of yield, size, zeta potential, morphology and protein content. EVs from differently treated LX-2 cells were isolated from the serum-free conditioned cell culture medium (CCM) after 24 h of treatment (CCMa, which includes treatment solutions) by differential centrifugation and ultracentrifugation (UC). Subsequently, they were purified by size exclusion chromatography (SEC) (see Methods). Cells were then washed with PBS and given fresh serum-free DMEM; EVs were isolated again after 24 h (CCMb). Unless otherwise stated, results herein stem from EVs isolated CCMb on the day of CCM harvest, even though the short-term stability of EVs was tested under different conditions for up to 21–28 days (Fig. S1). The full-size distribution profiles of the isolated EVs consistently showed polydisperse populations (Fig. 2a). Quantile subtraction of the distribution curve obtained from untreated cells showed that quiescent LX-2 produced larger EVs (>100 nm) more prominently than TGF-treated cells (Fig. 2b). Scanning electron microscopy (SEM) and transmission electron cryomicroscopy (cryo-TEM) imaging confirmed the polydispersity in the samples (Fig. 2c, d), as well as the expected morphology of the spherical, membrane-bound vesicles. After UC, up to 80% of total particles could be pelleted (Fig. S2). After SEC, EVs were successfully separated from protein aggregates co-purified during UC as determined by bicinchoninic acid (BCA) assay (Fig. 2e–g). Protein content associated with EVs was only detectable after 8 and 9 mL of elution upon SEC, and was comparable in all groups. Differently treated LX-2 yielded EVs in similar amounts, sizes, and zeta potential values (Table S1). Importantly, the documented cell viability at the time of CCMb collection was always above 95% (Table S1).

EV-containing pellets collected from CCMb from differently treated cells were analyzed by electrical/ asymmetric flow field flow fractionation (EAF4). AF4 technology allows for the unique separation of nanoparticles by two perpendicular flows[36,37]. It has been used with the aim of separating distinctive EV-subpopulations using in vitro and ex vivo samples[38,39], though not using HSC-EVs. The application of an electrical field in EAF4 approaches opens up the possibility of a deeper investigation based on electrophoretic mobility[40–42].

Fractionation of our EV-pellets after ultracentrifugation of CCMs reveals a broad size distribution of EVs around 300 nm with rather low electrophoretic mobility of $0.01 \times 10^{-8}$ m²/(V · s) and $1 \times 10^{-8}$ m²/(V · s) for DMEM, TGF and ROL/PA treated cells, respectively. In addition, only the CCMs of untreated and perpetuated cells exhibit another mixed population of larger EVs sterically eluting with smaller EVs (around 1 μm and 50 nm, respectively, electrophoretic mobility around $-7 \times 10^{-8}$ m²/(V · s)). While having similar elution times, there were differences in the emerged EV-subsets, especially by looking at the later eluting peak that was present in all samples (Fig. 2h–j). For ROL/PA, EVs in this second peak were generally larger compared to their DMEM and TGF counterparts, confirming the subtle size shifts observed when comparing EVs purified by SEC (Fig. 2a, b). Moreover, upon applying different electrical fields, a shift for the later eluting population of ROL/PA-EVs was noticed, whereas there was no shift noticeable in the fractograms of EVs from DMEM and TGF-treated cells. Statements regarding zeta potential are only semiquantitative, however, all fractograms of the EVs from the three treatment conditions of the producing cells exhibit a distinct fingerprint. Combined, this information points to the presence of unique EV-subsets, possibly relating to the phenotypical state of the cells from which they were isolated.

**Treatment of fresh LX-2 with EV-pellets isolated from differently treated LX-2.** We previously confirmed[29] that the combination of ROL/PA can deactivate LX-2 cells, and we reported how liposomes containing polyunsaturated phosphatidylcholines (PPCs) perform even better, as seen by the formation of

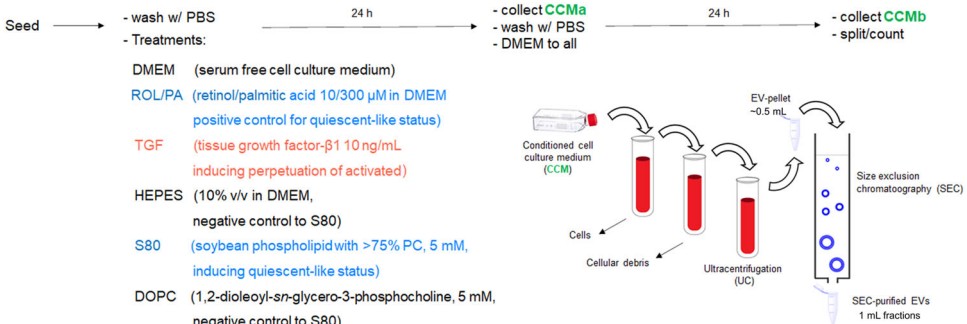

**Fig. 1 Workflow of EV isolation and purification protocols from LX-2 cells.** Schematic overview of the protocols to obtain highly purified EVs from LX-2 cells subjected to different treatments.

**Fig. 2 Isolation and characterization of EVs from LX-2 cells. a** Size distribution profiles of EVs isolated from differently treated cells (mean ± SD, $n = 3$). **b** Quantile subtraction of the yields. **c, d** SEM (**c**) and cryo-TEM (**d**) images of EVs isolated from untreated cells. **e–g** Protein content and vesicle number in the collected SEC-fractions obtained from untreated (**e**), quiescent (**f**), and perpetuated (**g**) LX-2 (mean ± SD, $n = 3$). **h–j** Representative EAF4 fractograms of EV collected from untreated (**h**), quiescent (**i**), and perpetuated (**j**).

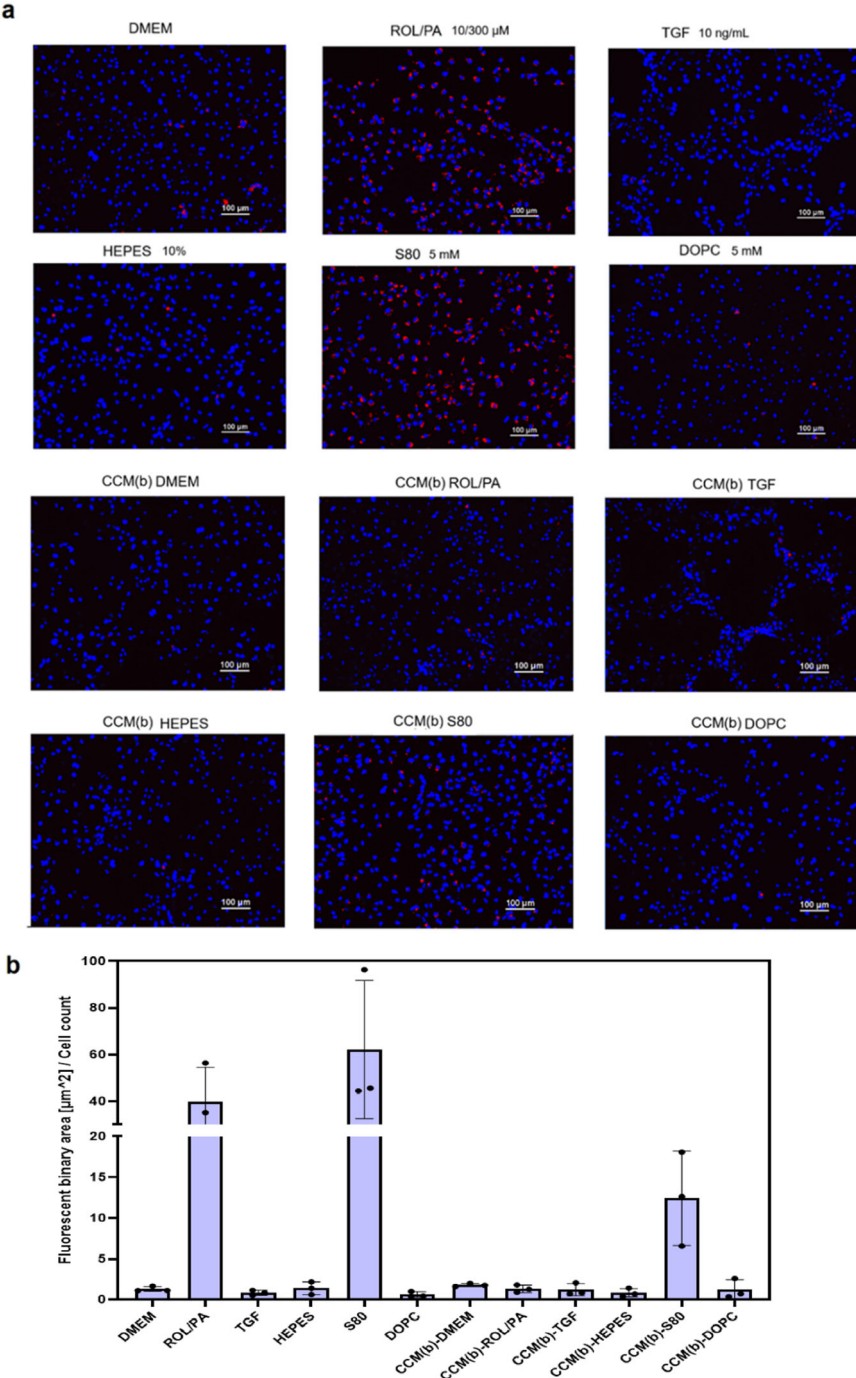

**Fig. 3 Treatment of fresh LX-2 with EV-pellets from previously treated cells. a** Representative images of ORO staining in fluorescence (seen as red spots; nuclei stained with blue DAPI) of differently treated cells after thresholding (see Fig. S3). **b** Quantitative analysis of stained lipid droplets, whereby the fluorescent area (correlating to a quiescent-like status) was normalized to cell count, and bars corresponding to treatment using EV-pellets are in purple (mean ± SD, $n = 3$). For more results, see Figs. S4–7.

cytoplasmic lipid droplets, which are indicative of the cells being in a quiescent-like status. Oil Red-O (ORO) staining was thus performed to reveal the presence of lipid droplets upon different treatments. Confirming our previous results, considerably more lipid droplets were identified in cells treated with S80 compared to any of the other treatments (Fig. 3a, b, for additional details, see Fig. S3). Using EV-pellets from CCMa (i.e., CCM that still includes treatment solution) resulted in trends mirroring those observed by direct treatment (Figs. S4, 7). This was true also when treatment using CCMa-EV was followed by treatment with

CCMb-EVs (Figs. S5, 7). While not as dramatic as the response to direct treatment, residual PPCs from CCMa cannot be excluded as a contributing factor. It is for this reason that the more striking results come from CCMb-EVs (Figs. 3a, b and S6, 7). Fluorescence microscopy images were quantitatively analyzed and used to evaluate cellular response (Fig. 3b). Cells that were treated with EV-containing pellets originating from the CCMb of TGF or S80-treated cells had a response that was similar to cells that were treated with TGF or S80 directly, albeit to a minor extent. When looking at the effects of CCMb-TGF on naive cells, the

microscopy images show that, remarkably, they arrange them-selves along a structured network, much like TGF-treated cells (Fig. 3a, panels TGF and CCM(b)-TGF). Functional spatial rearrangement of HSCs has indeed been previously reported[43]. Since the EVs were not purified by SEC in any of these cases, contamination with cellular factors co-purified during UC cannot be excluded. On the other hand, lipid droplets could still be found in cells treated using CCMb-S80 only, even though these cells were never in direct contact with the liposomal formulation, and even though any possible residual S80 from the supernatant of the treated LX-2 was washed away after the collection of CCMa. The newly stored lipids could be material that was recycled through EVs released by the originally treated LX-2.

Taken together, these results strongly suggest that EVs from either quiescent-like or perpetuated LX-2 cells might be sufficient to induce a correlated phenotypical status change in otherwise untreated cells.

**Proteomic analysis of SEC-purified EVs.** To further investigate the analytical differences that emerged from NTA, EAF4 (Fig. 2), as well as the biological effects exerted by EV-pellets (Fig. 3), we developed methods for the isolation and analysis of EV-associated proteins (see Fig. S8). We hypothesized that purified EVs allow correlation to the physiological state of their cell of origin. Mass spectrometry analysis was carried out on EVs purified by SEC using the peak fraction eluting from 7 to 8 mL (Fig. 2), leading to the identification of 3881 unique proteins.

A first examination of the SEC-purified EVs by principal component analysis (PCA) shows the degree of systematic variation in the proteomic profiles among biologically indepen-dent samples: EVs originating from similarly treated cells are more similar to each other than to the EVs from any of the other treatment setups (Fig. 4a). Hierarchal clustering of the same samples further demonstrated the similarities within treatment groups (Fig. 4b), as the unsupervised script correctly grouped protein profiles accordingly. The clustering of the proteins on the other axis of the heat map shows two more things. First, there are fundamental similarities across all HSC-EVs, with many protein families shared across samples, especially in the upper third of the heat map. Second, even with all these similarities, the rest of the heat map is characterized by differences in the intensity levels, and by patterns of missing protein hits indicated by white areas.

In a second exploratory step, we generated lists of proteins from all the single hits that would allow more immediate comparisons in a restrictive manner (see Supplementary Data 1). We decided to consider only proteins which were identified in all single replicates and which could be reliably quantified both by label-free quantification (LFQ) and by the sum of the three most intense peptide intensities (Top3) by MaxQuant, referring to them as persistent proteins from here on in. There were 3388 in total; out of those, 1931 proteins could be found and quantified in all replicates from all six different treatments (Fig. 4c). This data was mined to confirm the presence of established exosomal markers such as CD81 and CD9 tetraspanins in all samples, as well as the absence of known contaminants such as calnexin[27].

For every treatment group, there were proteins which were consistently found in addition to the 1931 proteins that were shared among all. A few of them were also exclusive, i.e., not strictly detected in any of the other groups (Fig. 4c). The thus generated lists of treatment group-specific persistent proteins were all cross-referenced against each other. A summary of the number of proteins found upon every direct comparison is found in Fig. 4d. Volcano plots for every comparison are found in Fig. S9.

Next, results from Welch's $t$-test were used to look for significant differences in protein recovery levels by group-wise

comparison of profiles from every condition. This created a new list of 1146 proteins that were either over or under-expressed in at least one of the single comparisons. Cross-referencing this list with the 1931 persistent proteins that were shared among all treatment groups yielded a panel of 44 proteins (Fig. 4e). For ease of comparison, a simple, normalized recovery score was developed by adding the LFQ values of every protein for each treatment condition and normalizing it to the sum of all of them, so that the panel could be visually inspected as a heat map (Fig. 4e). It can now be readily seen that TGF-EVs (negative control, indicative of perpetuated LX-2 cells) are more akin to DMEM-EVs than they are to ROL/PA-EVs (positive control, indicative of a quiescent-like status). What is even more remarkable, is that profibrotic EVs from TGF-treated cells are the dramatic, polar opposite of S80-EVs, in this rationally designed panel. This is not the case for EVs from DOPC-treated cells (negative PPC control to S80), nor for the EVs from the HEPES buffer control. This means that the observed effect is not a result of mere phospholipid treatment, but it stems from the combined benefit of specific bioactive, antifibrotic lipids present in S80. We have thus created a screening tool powerful enough to not only distinguish between our three basic controls (DMEM, ROL/PA, and TGF) but also hold the potential to semi-quantitatively evaluate the performance of additional treatments if further developed. An exploratory Gene Ontology (GO) analysis on the 44-proteins panel from Fig. 4e was performed using the Protein ANalysis THrough Evolutionary Relationships (PANTHER) platform[44,45]. There were no protein groups clearly associated with any specific pathway when taken together, only sparse hits (Fig. S10).

While such a panel of 44 proteins is considerably smaller than a full proteomic dataset, we postulated that there might be a selection of proteins to more simply tell apart quiescent-like LX-2 cells and their perpetuated (TGF-treated) counterparts by looking at the EVs they produced. There were 78 proteins persistently found in ROL/PA and S80 groups that were absent in the persistent protein profile of EVs originating from TGF-treated cells. Conversely, there were 4 proteins in the TGF group that were not consistently found in ROL/PA and S80 (Fig. 4f and Table S2). The thus selected proteins from our data were looked up by consulting the UniProt database[46] to find candidate protein markers within these two subsets that were either tissue-specific, membrane-bound, and/or secreted. Tissue specificity would be desirable for the translational applicability of our protocols, opening up the possibility of analyzing more complex ex vivo samples of EVs, while possibly being able to trace the EVs back to their origin[47]. We hypothesized that proteins that have been reported to be membrane-bound and/or secreted are less likely to be in the inner core of EVs. If so, they could be detected on the surface of EVs without destroying them. We have thus identified reduced lists of possible protein markers for our purposes from the ROL/PA and S80 subset (22 candidates), and from the TGF subset (4 candidate proteins).

**Proteomic analysis of AF4-purified EVs.** AF4 is a powerful technique increasingly adopted for the purification of nanosized particles, polymers, protein complexes, viruses, and even EVs[48–51]. During AF4 purification of our EV-samples, two main peaks emerged, and we collected both the early and late eluting peak (peak 1 and peak 2, respectively) (Fig. 2). Similar to the proteomic analysis performed for the SEC-purified EVs, AF4-purified samples were examined with the purpose of finding lists of treatment-discriminating proteins. We investigated peak 1 and peak 2 separately, as well as combined, leading to the identifi-cation of a total of 1807 distinct proteins.

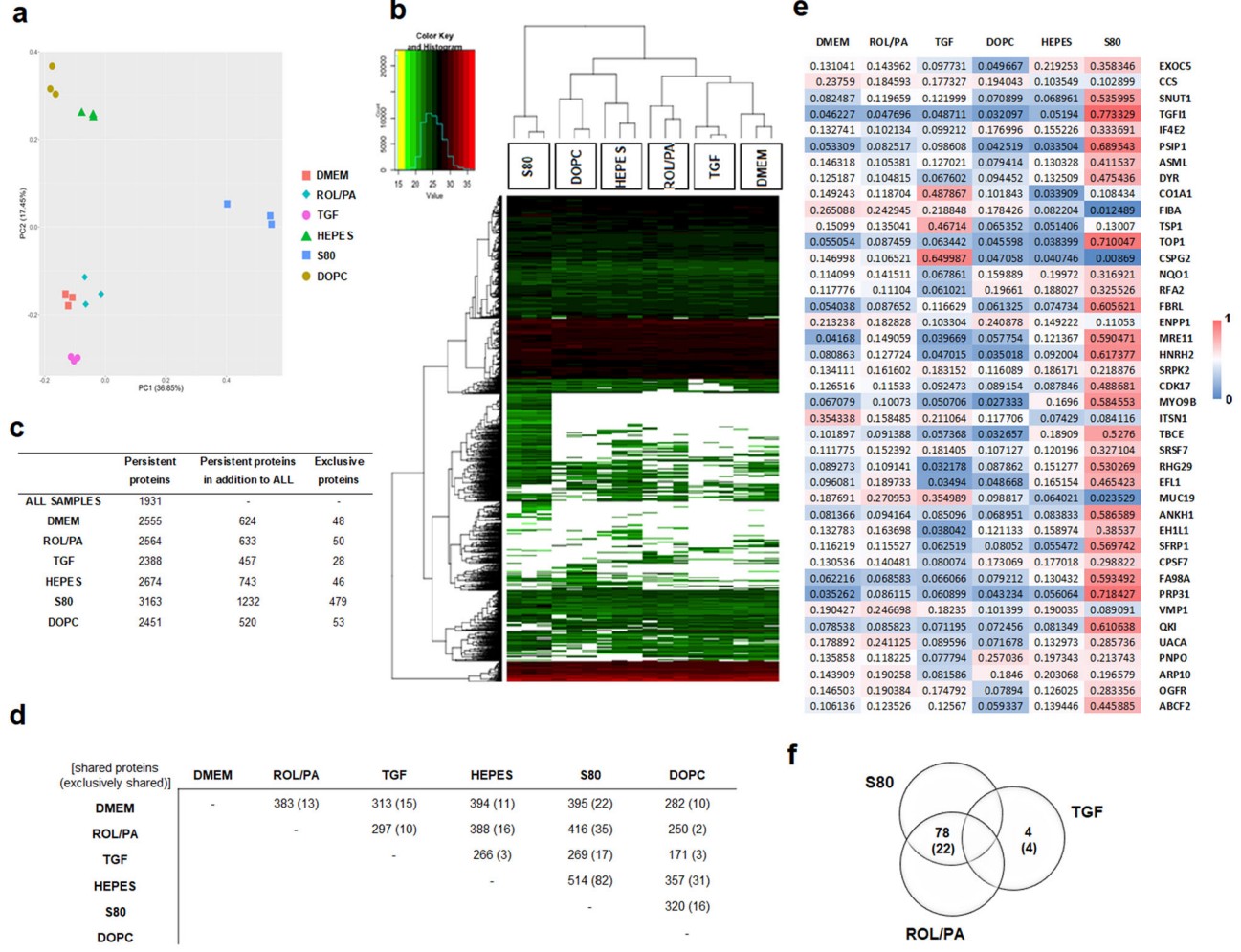

**Fig. 4 Main proteomic findings from SEC-purified EVs originating from differently treated LX-2. a, b** Principal component analysis and hierarchal clustering (also in Fig. S19) showing similarity degrees between biologically independent samples undergoing the same treatments, and differences between treatment groups. **c** Summary of the number of persistent proteins found in each treatment group. **d** Summary of the number of persistent proteins shared across treatment groups, excluding 1931 shared by all. **e** Panel resulting from cross-referencing the proteins consistently found in all treatment groups and the proteins that were under or over-expressed in the least one direct comparison after Welch's *t*-test. For ease of comparison, a simple, normalized recovery score was developed by adding the LFQ values of every protein for each treatment condition and normalizing it to the sum of all of them, so that the panel could be visually inspected as a heat map. **f** Venn diagram depicting the number of identified proteins persistently found in S80 and ROL/PA groups as opposed to TGF and vice versa; numbers in parenthesis indicate proteins within those subsets which were either tissue-specific, membrane-bound, and/or secreted. (Supplementary Data 1).

Compared to the SEC-purified EV protein profiles, AF4-purified EVs produced shorter lists. The explanation can be twofold. First, the SEC EV-containing peak was just one, and sharper than either of the collected AF4-peaks. This directly affects the precision with which the EV-associated proteins can be isolated, especially when considering that some could be found in both or either one of the AF4-peaks. Secondly, AF4-fractions are more diluted and limited in terms of recovered EVs, and, thus, in the recovered EV-associated proteins.

PCA revealed that peak 2 resulted in a better grouping of treatments compared to peak 1. This better grouping indicates that this particular fraction holds a distinctive EV subset that could be used for treatment discrimination (Fig. 5a–c). Combined analysis of AF4-peaks showed nonetheless a substantial number of proteins that can be found across both bands, resulting in mixed hierarchal clustering even when looking at peak 2 alone (Fig. 5d), and even after imputation (Fig. S11).

As with SEC-purified EVs, lists of treatment-correlating, persistent proteins were generated for AF4-purified EVs, both

by looking at the peaks separately and by looking at the peaks combined (Fig. 5e, f). Every treatment group was compared to each of the other ones: volcano plots for all the single comparisons are found in Figs. S12–14. The generated lists also allowed for a comparison of SEC and AF4 purification methods, summarized with Venn diagrams (Fig. 5g). The vast majority of the proteins found by AF4 purification were also detected in SEC samples. AF4-peak 2 had more original hits compared to AF4-peak 1, and indeed proteins isolated from AF4-peak 2 have greater overlap with SEC findings. Purification by AF4 led to the confirmation of many of the proteomic findings from SEC-purified EVs, split across two distinctive subpopulations of EVs, which would warrant further exploration in the future.

**Fluorescence NTA method for the detection of EV-associated proteins correlating to cell status.** The detection of specific proteins in EVs is an integral part of the field. Typically, western blots is the method of choice to show the presence of known

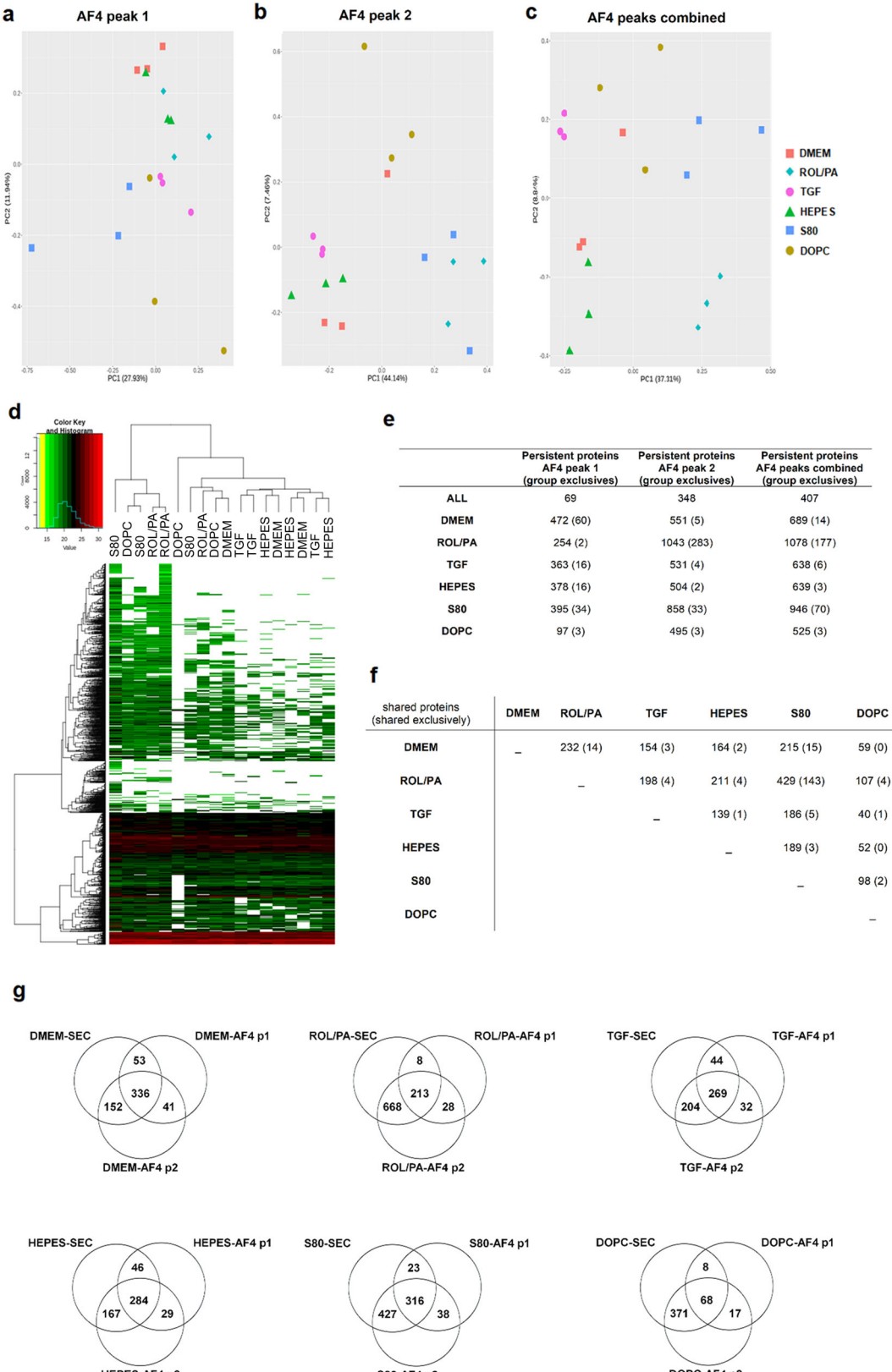

**Fig. 5 Main proteomic findings from AF4-purified EVs originating from differently treated LX-2. a–c** Principal component analysis for samples originating from AF4-peak 1 (**a**), peak 2 (**b**), and for the two peaks combined (**c**). **d** Hierarchal clustering for biologically independent AF4-peak 2 samples undergoing the same treatments. **e** Summary of the number of persistent proteins found in each treatment group. **f** Summary of the number of persistent proteins shared across treatment groups. **g** Venn diagram comparisons of SEC and AF4-purified samples: for every treatment group, the proteins found in the SEC fraction are compared to those found in AF4-peak 1 (AF4 p1) and AF4-peak 2 (AF4 p2).

exosomal markers in EV-samples, including CD9, CD81, CD63, and Alix[52–58]. Alternatively, strategies relying on flow cytometry (FACS) are employed. Using FACS when working with EVs can be challenging, given that the instruments were designed for the analysis of single cells, which are considerably larger in size[59]. A workaround for this particular issue is the coupling of EVs to large beads, and subsequently analyzing the presence of the markers of interest on the beads covered in EVs. Nano-flow cytometers have recently entered the market for the specific analysis of nanoparticles in a way that is analogous to how classical FACS instruments work with full cells[60–63]. Research groups have been trying to measure fluorescently labeled EVs directly by fluorescence nanoparticle-tracking analysis (f-NTA), demonstrating the feasibility of its application[56,64,65]. These reports used unspecific dyes or immunolabeling for specific proteins, often opting for quantum dots (QD) conjugated antibodies to overcome photostability problems associated with many conventional fluorophores[56]. The latter approach, however, can be vitiated if effective protocols to purify QD-labeled EVs from free QD are not effectively validated.

Our proteomic analysis of EVs has yielded a list of candidate protein markers, which could be detected on the surface of EVs without destroying them, thus allowing for their analysis by f-NTA. The secreted protein acidic and cysteine-rich (SPARC)[31,32,42,66] was thus selected as a proof-of-concept protein, since it was one of the four proteins consistently found in TGF samples, but not in ROL/PA and S80 (Fig. 4f). We demonstrated that, while not necessarily an integrated part of EV-membranes, SPARC was associated strongly enough to EVs to be co-purified after SEC and AF4. SPARC is also reportedly secreted or found in the extracellular region, in or around the basement membrane, indicating that when found in EV-samples, it is more likely to be associated with their membrane than with their aqueous inner compartment. Importantly, SPARC can be highly expressed in tissues undergoing wound repair or morphogenesis[31,67,68], including the liver[69,70], making it overall an excellent candidate for reporting on the physiological state of the cells that released the EVs with which SPARC could be found.

To develop immunolabeling methods for the non-destructive analysis of physiologically relevant EV-associated proteins by f-NTA, we first assessed the feasibility of detecting fluorescently labeled EVs in general. PKH dyes are used in a wide variety of instances for the nonspecific labeling of cellular and vesicular membranes[71–73]. SEC-purified EVs from untreated LX-2 cells were successfully labeled with the PKH67 membrane dye[74]. Testing different concentrations over time, we could determine that the labeling plateaued after 15 min (Fig. S15). We settled on 20 min of incubation time when we systematically increased dye concentration, showing that almost all of the detected particles were indeed membranous, and that an almost linear dose-dependency could be reliably measured ($R^2 = 0.909$, Fig. 6a). The use of PKH dyes has been recently called into question, particularly because of their hydrophobicity leading to the formation of dye aggregates that can be detected by NTA, significantly affecting the size distribution profile[75,76]. However, given the freshly prepared PKH67 concentrations that we used in our samples with relatively few EVs (Table S1), we did not encounter new subpopulations of nanoparticles in our measurements compared to unstained samples (for representative size distribution profiles of PKH67 labeled particles see Fig. S16).

The next steps involved labeling EVs with an AlexaFluor488-conjugated secondary antibody. We chose to start with CD81, since we found it to be present in all EV-samples in our proteomic analyses. Being a tetraspanin, it was also likely to be available for binding without destroying the EVs, and previous reports have shown that it could be detected by f-NTA[56,64].

The first approach was to incubate AF488-CD81 directly in the SEC-purified EV-samples originating from DMEM-treated LX-2 cells, at different dilutions of AF488-CD81, for various amounts of time and at different temperatures. Higher concentrations increased the number of detected particles in fluorescence mode, but not in a dose-dependent manner (Fig. 6b). Incubation at 37 °C was not significantly better than incubation at room temperature, although fluorescent particles were detectable starting from an earlier time point (Fig. S17). Incubation time had a consistently higher influence and we found the range between 4 and 6 h optimal for our samples. Longer times (18 h) could be tested at lower temperatures (4 °C) in order to split the workload of EV-isolation and EV-analysis, while minimizing EV-loss.

Without a purification step for excess AF488-CD81 prior to measurement, the background intensity was always too high for the software, regardless of final dye dilution. Our relatively low EV-yields resulted in limited final dilutions of the SEC-fractions before NTA measurements. More concentrated EV-samples could be labeled after SEC with fewer issues.

To obviate these limitations, we decided to incubate AF488-CD81 with the EV-pellet resuspended after UC, right before the SEC purification we perform regardless. The biggest drawback of this incubation strategy is the limited number of EV-pellets that can be obtained in one day; additionally, it slows down the already time-limiting SEC step. There was no significant difference between 3 and 5 h incubation times, other than the smaller standard deviations for the latter instance, which is why we chose it for further experiments (Fig. 6c).

Since 8 ng/mL resulted in higher labeling, that concentration was used when we looked at CD81 and CD9 labeling both separately and combined (Fig. 7d). While we can neither confirm nor exclude co-localization of the two markers on a single EV, the combined results suggest that there is some incomplete overlap. This means that double or even triple staining with exosomal markers might be a viable strategy to cover 100% of the particles detected in scatter mode for a sample of pure exomes. Antibody concentrations for incubations with EV-pellets could be further optimized to account for EV-abundance in the samples.

As a final step, we compared our newly developed f-NTA methods to the analysis of EV-markers by FACS using 6 ng/mL of AF488-CD9 (Fig. 6e). Both methods detected CD9 positive events in all of the samples- irrespective of cell treatment prior to EV harvest—which validated. the presence of exosomal markers in our EV populations by all three proteomics, FACS, and f-NTA. FACS analysis resulted in higher percentages of fluorescently labeled events (up to 70%) compared to f-NTA (around 30%), but also with considerably higher standard deviations. Our f-NTA methods, on the other hand, performed more reliably, especially considering that there were no false positives, i.e., no particles could be detected in samples incubated with isotype controls.

Our optimized immunolabeling methods for the detection of proteins on single EVs by f-NTA could be transferred to check for the presence of SPARC on EVs isolated from differently treated LX-2 cells. Remarkably, f-NTA measurements were consistent with proteomic findings. The presence of CD81 was consistent across samples, and the presence of SPARC on EVs was reproducibly different for different treatment groups, with HEPES and TGF samples having the highest amounts, and S80 especially having hardly any (Fig. 6f). We thus found that S80 greatly reduces the relative presence of SPARC-positive EVs because it either actively suppress it, or its mechanism of actions results in its lowered expression. SPARC presence in EV-samples from DOPC-treated cells (less than 10%) was also decidedly lower than for HEPES (70%) and TGF (40%) in our f-NTA measurements.

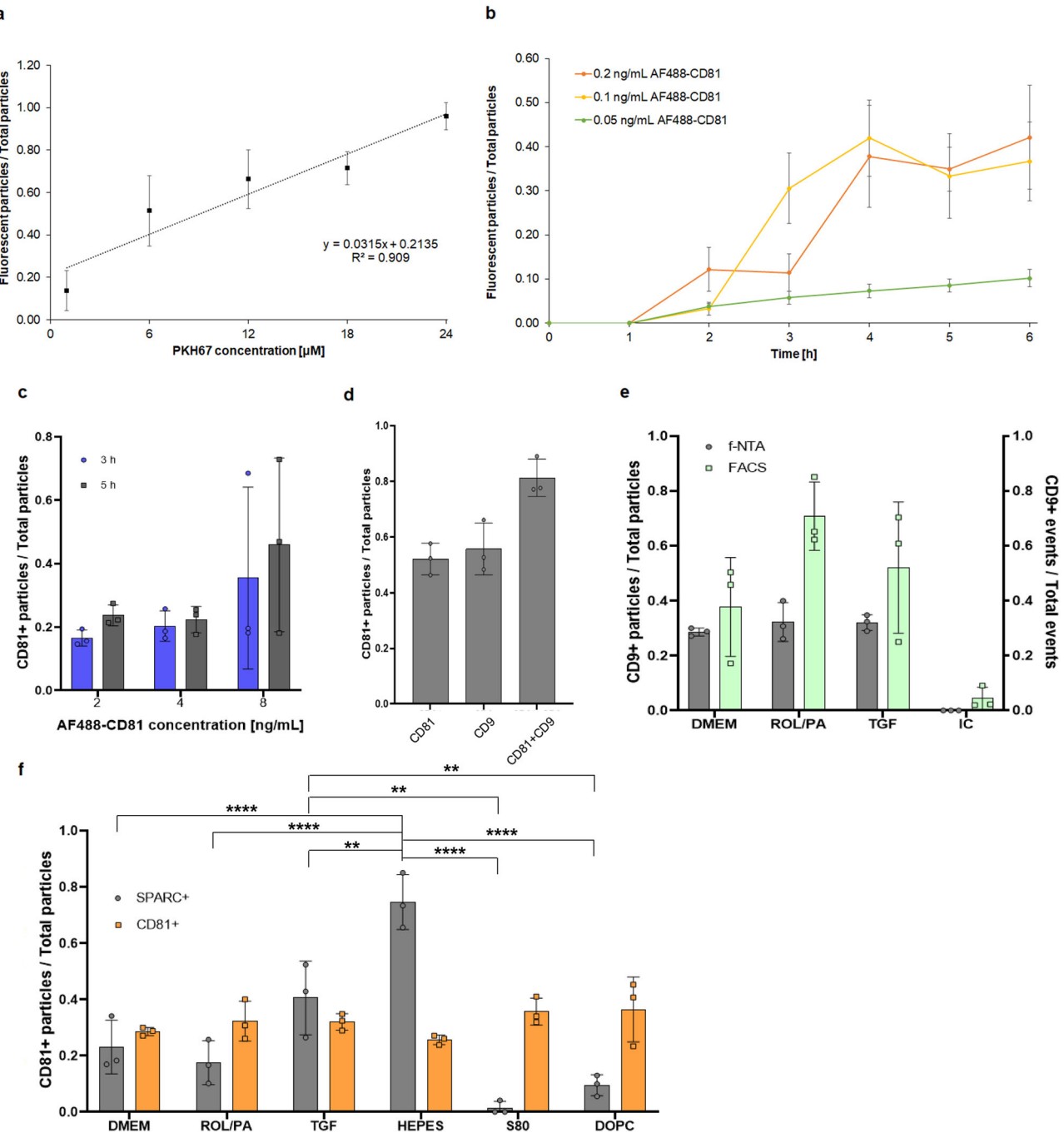

**Fig. 6 Feasibility of f-NTA for the detection of physiologically relevant proteins. a** SEC-purified EVs stained with unspecific membrane dye PKH67 by systematically increasing dye concentrations. **b** SEC-purified EVs incubation with varying amounts of AF488-CD81 and for different times at 24 °C. **c, d** EV-containing pellets incubated with AF488-CD81 prior to SEC and detected after purification (**c**) and comparison with additional incubation with AF488-CD9 (**d**). **e** Direct comparison of CD9 detection by f-NTA and FACS using EVs from differently treated LX-2, including isotype control (IC). **f** Detection of SPARC and CD81 on EVs isolated from differently treated LX-2 cells following the optimized f-NTA protocol (mean ± SD, $n = 3$). P values ($p \leq 0.05$ (*), $p \leq 0.01$ (**), $p \leq 0.001$ (***), and $p \leq 0.0001$ (****)) were determined by one-way ANOVA on ranks and Tukey's multiple comparison.

We do not know by which mechanism is SPARC being incorporated into EVs. The higher presence in some of our samples compared to others could be directly linked to an increased expression of the protein. It could also be due to an improved affinity to EV-surface, either because of treatment-induced changes in EV-composition (see Fig. 4) or because of treatment-induced differences in the extracellular chemical environment, which might lead to a worsening of SPARC's affinity to the extracellular matrix for example.

Nevertheless, these results prove that f-NTA could be used for a quantitatively meaningful detection of physiologically relevant EV-associated proteins by incorporating the incubation of secondary antibodies into rigorously established EV-isolation and purification steps. This ultimately provides a convenient alternative to more time-consuming and hands-on heavy western blot and flow cytometry protocols. Moreover, we could measurably correlate the cellular response to PPC treatment such as S80 to the lowering of the relative presence

of SPARC on the generated EVs, offering insights into their effectiveness.

## Discussion

We reported on highly reproducible isolation and purification methods for EVs released by LX-2 HSCs upon six different treatments, affecting the LX-2 phenotype in different ways and to different extents. The heterogeneity of EV-samples has hindered the research into distinctive subpopulations due to their overlapping sizes, densities, zeta potential values, and biochemical compositions[39]. We were nonetheless able to observe subtle size shifts within these polydisperse samples of EVs originating from differently treated cells by NTA, as well as prominent dissimilarities in EAF4. The groundwork EV-analysis optimized herein begs for further methods of characterization, such as the inspection of RNA or glycosylation patterns. Nonetheless, our EV-analysis for an immortalized cell line used as widely as LX-2 is pivotal to boost more investigations into possible antifibrotic treatments by evaluating the resulting EVs.

When used to treat fresh cells, EV-pellets from differently treated cells were sufficient to induce a physiological response similar to direct treatment, as observed by fluorescence microscopy. However, the investigation into the biological role of EVs is also far from being exhausted. It would be important to see for how long after treatment do HSCs preserve the phenotypical changes acquired. Importantly, we do not know for how long would their EVs be effective in causing those same phenotypical changes onto naïve cells, and how do the protein profiles of the latter's EVs look like.

Notably, EV-lipid composition for different classes varied depending on the original treatment, also displaying selective enrichment of biomolecules compared to cells. SEC and AF4 purification methods showed astounding similarities in the recovered EVs' lipid profiles.

Proteomic data of SEC and AF4-purified EVs was mined to find candidate protein markers to discriminate EV-samples between differently treated HSCs. Although beyond the scope of this particular study, our proteomic data can be further examined by detailed gene set enrichment analysis (e.g., network and pathway analyses) for all the generated protein lists. We summarized in our tables more than 120 distinctive lists of proteins, all interesting in their own right. Deep learning approaches could be used to sort through different EV-proteomic data to find the minimum, specific signature profile for HSCs, similarly to what had been recently done for the identification of EVs originating from cancer cells[58].

SPARC was rationally selected to successfully explore the feasibility of using f-NTA as a non-destructive tool for the aforementioned discriminatory exercise. By thus looking into the relative presence of SPARC-positive EVs, we found that S80 greatly reduces it. Our optimized f-NTA method could be applied in the future to EVs from LX-2 upon even more different treatments, effectively screening drug candidates targeting fibrogenesis. While SPARC proved to be an excellent marker for negative impact in our model, having a working candidate protein to report on positive phenotypes would be all the more meaningful, especially with a side-by-side comparison. We have tried selecting proteins from the quiescent-like status list (Fig. 6f) which could be detected non-destructively, namely GPC1 and IKKB, but we were unable to detect effective immunolabeling with either one of them by f-NTA. It is possible that the labeling might need protein-specific optimization steps, or that proteins were not on the surface of the EVs, requiring perhaps a permeabilization step in the protocol.

However, there are at least almost 20 more interesting candidates to test from our data. The method could be expanded to include more EV-associated, clinically relevant proteins, even within the same sample. NTA systems with more than one laser and fluorescence channel would allow for the almost simultaneous analysis of multiple proteins, provided they are labeled with non-interfering dyes. While co-localization of single proteins onto single EVs would not be directly measurable by NTA yet, their collective presence within the same EV-samples is already technologically possible to verify.

Our results pave the way for more precise clinical analyses: markers related to diseased and healthy states, as well as proteins that are tissue-specific or preferentially expressed by specific cells, could all be conceivably checked within one EV sample if appropriately selected. Assessing the translational applicability of our protocols for the evaluation of EVs from primary cells, or blood from healthy volunteers or patients would be the most important perspective validation to accomplish.

## Methods

**Lipid vesicles preparation**. Liposomal formulations with soybean phospholipids with 75% phosphatidylcholine (S80, kindly donated by Lipoid GmbH) or pure phospholipids 1,2-dioleoyl-sn-glycero-3-phosphocholine (DOPC, Lipoid) were prepared by thin film hydration as previously described[29]. Briefly, extruded 10 times through a 0.2 μm polycarbonate membrane. The hydrodynamic diameter and the size distribution (polydispersity index, PDI) of the liposomes were measured with a Litesizer 500 (Anton Paar).

**Cell culture**. LX-2 cells were grown in high glucose (4500 mg/L) DMEM (Carl Roth) supplemented with 200 mM L-glutamine (Sigma), 10,000 units/L of penicillin and streptomycin (Gibco), and 2% (v/v) of sterile filtered (0.2 μm, cellulose acetate membrane) fetal bovine serum (FBS, Merck Millipore).

For experiments, $1.2 \times 10^6$ LX-2 were seeded in T175 cell culture flasks and cultured for 120 h, or $1 \times 10^5$ cells/well were seeded in 12-well microtiter plates and cultured for 24 h. Cells were then washed with phosphate-buffered saline (PBS) and treated for 24 h with different solutions prepared in serum-free cell culture media (DMEM): either ROL/PA (10/300 μM), TGF (10 ng/mL, 227.27 pM), or liposomal formulations (5 mM lipid concentration) of S80 or DOPC.

**EV-isolation and purification**. LX-2 cells were treated in serum-free conditions for 24 h and the CCM from two T175 flasks per treatment ($2 \times 25$ mL) was collected (CCMa, which includes treatment solutions). Cells were then washed with PBS, and, regardless of previous treatment, they were all supplied with fresh serum-free medium (DMEM). After 24 h, the CCM was collected again (CCMb) and cells were split to determine their number and viability. CCMs were centrifuged (300×g, 3 min, 4 °C), the supernatant was moved into a new tube and centrifuged again (9000×g, 30 min, 4 °C). The pelleted cell debris was discarded and the supernatant was ultracentrifuged (120,000×g, 2 h 30 min, 4 °C, Beckman Coulter, Optima XPN Ultracentrifuge, Type 70 Ti rotor or SW 32 Ti rotor). After discarding the supernatant, the EV-containing pellet was resuspended in 0.5 mL of PBS and purified by size exclusion chromatography (Sepharose CL-2B).

All of this is summarized in Fig. 1.

The collected SEC-fractions were analyzed for protein content by means of bicinchoninic acid (BCA) assay. Particle yield, size distribution profiles, and zeta potential were determined by nanoparticle-tracking analysis (NTA, ZetaView 8.05.05 SP2 equipped with a 488 nm laser, zeta potential and temperature control units, and Particle Metrix). Measurements were performed at 25 °C, a camera sensitivity of 80, and a 100 ms$^{-1}$ shutter value. Particles were traced for at least 15 consecutive frames, videos were taken at 11 positions. Samples had to have ≥200 traced particles.

Results from the single SEC-fractions were consolidated to obtain an average yield and an average particle size, as well as a combined size distribution profile for EVs originating from the differently treated LX-2. Unless otherwise stated, the presented results refer to EVs isolated from CCMb. Every sample was freshly analyzed on the day it was collected, except for those used for electron microscopy imaging, which had been previously stored at −80 °C[51,77].

EV-yield and size distribution profiles have been checked on EVs stored for up to 21 days under different conditions: −80, −25, 4, and 37 °C and at room temperature after being freeze-dried with 1% trehalose (w/v)[77].

**SEM and cryo-TEM**. For scanning electron microscopy (SEM), 10 μL resuspended EV-samples were left to dry overnight on silica wafers; they were then sputter-coated with a thin layer of gold for imaging under high vacuum with an accelerating voltage of 5 kV using a Zeiss EVO (Zeiss EVO MA15 LaB6, Oberkochen,

Germany) instrument. SEM was performed with the help of Dr. Chiara De Rossi (Helmholtz Institute for Pharmaceutical Research, Saarbrücken).

For cryogenic transmission electron microscopy (cryo-TEM), samples were prepared as previously described[78]. Briefly, 5 µL of EV sample were transferred to a copper grid covered by holey carbon film (R1/2, 300 mesh, Quantifoil Micro Tools, Großlöbichau, Germany) and excess liquid was blotted between two strips of filter paper. Samples were plunged into liquid ethane (180 °C) in a cryobox and they were rapidly moved with a Gatan 626 cryo-transfer holder into the pre-cooled cryo-electron microscope (Philips CM 120, Munich, Germany) operated at 120 kV. The Images were acquired with a 2k CMOS Camera. Cryo-TEM imaging was performed by Dr. Jana Tamm (Friedrich Schiller University, Jena).

**EAF4**. EV-pellets from ultracentrifugation were resuspended in ~550 µL of DMEM. Volumes of 100 µL were injected from a pump and autosampler (Agilent Technologies Germany, Waldbronn, Germany), sample tray cooled to 8 °C, with an Eclipse Dualtec (Wyatt Technologies Europe, Dernbach, Germany) equipped with a Mobility EAF4 (Wyatt), a UV absorbance detector (Agilent), and a multi-angle light scattering (MALS) detector Dawn Heleos (Wyatt) for particle detection and size measurement. The Mobility channel was prepared with a narrow spacer (250 µm) and contained a 30 kDa molecular weight cut-off regenerated cellulose membrane, which was equilibrated with six injections of cell culture supernatant with 10 mM phosphate buffer pH 7.4 as mobile phase and a detector flow rate of 1 mL/min. After equilibration of 1 min in focus mode with 1.5 mL cross-flow, the sample was injected in focus mode for 5 min, then eluted for 20 min at 0.2 mL cross-flow followed by a linear decrease over 5 min to 0.03 mL cross-flow and held for 10 min. This was followed by a washout phase at 0 mL/min cross-flow and an elution inject step. Different amperages ranging from +2 to −6 mA were applied during the elution with cross-flow phase in consecutive runs. For pre-parative fractionation, the same hardware and membrane were used with a narrow spacer (350 µm) and the focus inject time was increased to 8 min to accommodate and focus the injection volume of 500 µL in the channel. Samples were collected at 1 mL per fraction with an automated fraction collector (Agilent), which was set to the respective sample peaks. Collection times for peaks 1 and 2 were as follows: 14.5–18.5 min and 33.5–38.5 min for DMEM, TGF, and DOPC, 21.5–25.5 min and 34.5–42.5 min for ROLPA, 15.5–19.5 min and 34.5–42.5 min for HEPES, and 13.5–19.5 min and 30.5–37.5 min for S80-EVs.

**Treatment of fresh LX-2 with EVs isolated from differently treated LX-2**. Extracellular vesicles (EVs) produced by LX-2 cells in T175 flasks were isolated from serum-free conditioned cell culture medium (CCM) after 24 h of treatment (CCMa) by differential centrifugation followed by an ultracentrifugation step (see above). Cells were then washed with PBS and given fresh serum-free DMEM. EVs were isolated again after 24 h (CCMb). The whole EV-containing pellets originating from CCMa and CCMb were directly resuspended after UC in fresh, serum-free DMEM, and they were then used to treat LX-2 cells seeded in 12-well plates the day before for 24 h. The presence of cytosolic lipid droplets was determined by Oil Red-O (ORO) staining as previously reported and briefly described below.

**Analysis of lipid droplet content—ORO staining**. After cell treatment, LX-2 in 12-well plates were washed with PBS, fixed with Roti®-Histofix, and stained with a 0.5% (w/v) ORO solution in propylene glycol. Nuclei were counterstained with DAPI. Fluorescence and phase contrast image acquisition was performed using a Nikon Ti-U inverted microscope. The quantification of the ORO-stained, fluorescent binary area was normalized to the cell count as determined by DAPI-stained nuclei within the image after thresholding (Fig. S3). For every condition, a total of at least 27 images were acquired: three images/well, from three separate wells, repeated with three biologically independent replicates (performed on separate days from different cellular splits).

**Proteomic profiling of EVs**. SEC-purified EVs, as well as AF4-fractioned samples, were transferred to a pre-conditioned polyvinylidene fluoride (PVDF) membrane. First, the PVDF membrane was cut into uniform disks, and the pieces were wetted with methanol (MeOH) for 5 min. After removal of the MeOH, a solution of 0.05% sodium dodecyl sulfate (SDS)/5% MeOH/0.05% Dithiothreitol (DTT) was added for 5 min. Membrane pieces were placed into the bottom of the collection tubes right before EV collection with a little PBS to keep them wet. After the purification step (either by SEC or AF4), the PVDF membranes and EV-containing samples were centrifuged (3000×g, 1 h, RT), and the supernatant was discarded. PVDF membrane pieces were dried under N₂ flow for 15 min and stored at 4 °C. Comparative, shotgun proteomics was performed after reductive alkylation and trypsin digestion of the samples[79,80] by the Proteomics Mass Spectrometry Core Facility (PMSCF) at the Department of Biomedical Research (DBMR) of the University of Bern. Peptides were analyzed by liquid chromatography-tandem mass spectrometry (nano-LC-MS/MS) and spectra were searched by MaxQuant/Andromeda[81–83]. The mass spectrometry proteomics data have been deposited to the ProteomeXchange Consortium via the PRIDE[84] partner repository with the dataset identifier PXD037453.

**Staining of EVs with PKH67**. SEC-purified EVs were incubated with 1.5–24 µM of PKH67 (MilliporeSigma) for 5–120 min at 24 °C under gentle shaking. NTA measurements were performed in scatter mode as previously described (see above), but the sensitivity was changed to 90 for measurements in fluorescence mode (f-NTA).

**Detection of exosomal marker CD9 by flow cytometry**. The analysis of EVs by flow cytometry (FACS) was performed as previously reported[85]. Briefly, EVs were coupled to 4 µm aldehyde/sulfate latex beads. EV-containing SEC-fractions (1 mL each, see above) were divided into 0.5 mL aliquots. Freshly filtered (CA, 200 nm) BSA 1% w/v in PBS was also prepared as a negative control. All samples were then incubated for 15 min at RT with 10 µL of latex beads. PBS was added up to 1 mL and all samples were incubated for 1 h at RT with gentle shaking. The reaction was stopped with 0.5 mL of 200 mM glycine and incubated for 30 min at RT. Beads coupled to EVs (or BSA) were centrifuged (2000×g, 3 min, RT), the supernatant was removed, and the pelleted beads were resuspended with BSA 1% w/v in PBS. This washing step was repeated two more times. Samples were stained with fluorescently labeled antibodies for CD9 (6 ng/mL, IgG2b, Clone #209306) or with the fluorescently labeled isotype control (IC) in ice and in the dark for 30 min. Finally, samples were washed twice with BSA 1% in PBS and analyzed on a BD LRS Fortessa (BD Biosciences) using BD FACSDiva 8.0.

**Antibody labeling**. Antibodies against human CD81 (IgG2B, Clone #454720, Biotechne) and SPARC (IgG1, Clone #122511, Biotechne) were conjugated with AlexaFluor488 (AF488) using the Lighting-Link® (LL) antibody labeling kit (Biotechne) as per manufacturer's instructions and under sterile conditions. Briefly, the LL-modifier solution was added to the unconjugated antibody in sterile PBS (1 µL for every 10 µL of antibody). This was then used to re-suspend the lyophilized mixture with AF488 and left incubating for 15 min at RT. LL-quencher was then added to the antibody-AF488 mixture (1 µL for every 10 µL of antibody). The LL-kit was also used to prepare an isotype control with the IgG chain (Goat anti-Human IgG (H + L), Cross-Adsorbed Secondary Antibody, Thermo Fisher Scientific). The performance of EVs of AF488-CD81 already conjugated at purchase and LL-conjugated AF488-CD81 were compared as well (Fig. S18).

**Detection of EV-associated proteins with f-NTA**. Incubation with AlexaFluor488-conjugated antibodies for exosomal marker CD81 (AF488-CD81) was performed directly into SEC-purified samples at different times (every 10 min up to 120 min, every hour for up to 6 h and overnight), at different temperatures (in ice, at 24 and 37 °C), with different concentrations (0.05–1 ng/mL). Only data with successful f-NTA detection could be shown.

Incubation with AF488-CD81, LL-AF488-CD81, AF488-CD9, and LL-AF488-SPARC, were performed in the resuspended EV-containing pellets obtained after UC, prior to SEC. The protocol optimization was done with AF488-CD81. The tested conditions included different incubation temperatures (24 and 37 °C), for variable amounts of time (for 3 or 5 h), using 2–8 ng/mL of antibody conjugates.

Measurements were performed in scatter mode as previously described (see above), but the sensitivity was changed to 90 for measurements in fluorescence mode. For every protocol yielding a measurable result, incubation with isotype controls (either bought already conjugated or LL-conjugated) were also performed.

For antibody incubations with EV-pellets originating from differently treated LX-2 cells, the regimens were as follows: DMEM, ROL/PA (10/300 µM), TGF (10 ng/mL), HEPES buffer (10% v/v), S80 (5 mM), and DOPC (5 mM). The labeled EVs were then purified and collected after SEC.

**Statistics and reproducibility**. Data, when applicable, are presented as mean ± standard deviation (SD) from at least three independent samples unless otherwise indicated. For multiple group comparisons (Fig. 5f), statistical differences were assessed with a one-way ANOVA test combined with Tukey's multiple comparisons test after performing a normality test (Shapiro–Wilk) using GraphPad Prism version 9.4.0 for macOS, GraphPad Software, San Diego, California USA, www.graphpad.com. Results were considered statistically significant if $p \leq 0.05$ (*), $p \leq 0.01$ (**), $p \leq 0.001$ (***), and $p \leq 0.0001$ (****).

**Reporting summary**. Further information on research design is available in the Nature Research Reporting Summary linked to this article.

## Data availability

All data generated or analyzed during this study are included in this published article and its supplementary information files. Proteomics data were available as Supplementary Data 1 and via ProteomeXchange with identifier PXD037453. Raw data of Figs. 2, 3, and 6 are available as Supplementary Data 2.

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

## Acknowledgements

The Phospholipid Research Center is kindly acknowledged for the financial support (Grant ID: PLU-2017-056/2-1). C.Z. and P.L. would like to thank the staff of the Proteomics and Mass Spectrometry Core Facility (PMSCF), Department for BioMedical Research (DBMR), University of Bern, Switzerland. The morphology of the isolated EVs was evaluated under scanning electron microscopy (SEM) with the help of Dr. Chiara De Rossi (Helmholtz Institute for Pharmaceutical Research, Saarbrücken). Cryogenic transmission electron microscopy (cryo-TEM) imaging was performed with the help of Dr. Jana Stamm (Friedrich Schiller University, Jena). G.F. would like to acknowledge financial support from the NanoMatFutur program from the Federal Ministry of Research and Education (Grant number 13XP5029A).

## Author contributions

P.L. and G.F. conceived the original project and supervised it. C.Z. devised and performed all the experiments and analyzed all data. K.F. devised, carried out, and analyzed the EAF4 experiments. C.Z. led manuscript writing. P.L. and G.F. interpreted data and revised the manuscript. All authors contributed to the discussion.

## Competing interests

No private study sponsors had any involvement in the study design, data collection, or interpretation of data presented in this manuscript. P.L. declares the following competing interests: she has consulted Lipoid GmbH and Sanofi-Aventis Deutschland and received research grants from Lipoid, Sanofi-Aventis Deutschland and DSM Nutritional Products Ltd. C.Z., K.F., and G.F. declare no competing interests.
