## [Peer Review File · Communications Biology]

Reviewers' comments:

Reviewer #1 did not provide a full review but emailed some minor comments:

Minor comments to the authors

1) In the M&M section (EV isolation and purification) despite the authors nicely described the protocol, final details regarding volume of EV-pellet resuspension or SEC pooled fractions are missing. These data are necessary to understand the amount of particles used for SEM or cryo-TEM as well as extremely useful for the replication of the results by other groups. They may also serve as a starting reference for other related-projects. This data could be included in the Scheme figure 1.

2) In the M&M section (Treatment of fresh LX-2 with EVs isolated from differentially treated LX-2), data regarding particle dose/volume dose are missing. These data are important for future replication of the results and reference points for other researchers. Did the authors try different EV particle concentration? Please include this information and clarify if dose-dependent effects were observed.

3) In the M&M section (Staining of EVs with PKH67), authors do not give details regarding potential removal of dye (washing steps) or any other considerations, especially important considering the controversial opinion about PKH67 results in the field.

4) In the Results section, "EV isolation, purification and characterisation", Fig. S2 shows a "change on UC protocol" on panel b. However, there is no mention about these changes in the manuscript. Could the authors clarify these experiments? New UC protocol may be the described one on M&M section, but, what was the "worse" one? It may be quite useful for other researchers in the field.

5) In the Results section "Treatment of fresh LX-2 with EVs isolated from differentially treated LX-2", it would be nice to refer to Fig. S3 directly on the text as well as on the Fig.2 caption. In addition, the authors claimed on the text "striking results from TGF treatment", however, when Figure 2 is checked, TGF results do not seem so striking and if so, the results may be related to ROL/PA and not TGF. Please clarify.

6) In the same Results section, authors described a "structured network" but this is not easily appreciable on Figure 2 (or any other figure). May bright-field images help to this aim? Could the authors implement the Figure+panel references on the text? Since abbreviations are allowed, it may help the reader and not elongate too much the text.

7) Figure 2 results may be improved by breaking the Y-axis on panel Y to emphasise those results at lower fluorescence ratios. Connection to Supplementary Figure results maybe useful to understand.

Reviewer #2 (Remarks to the Author):

Zivko et al present an interesting work in the development of a fluorescence-based NTA assay to identify secreted proteins that may be potentially used as non-invasive biomarkers of liver fibrogenesis. The authors characterized EVs isolated from cells undergoing the various treatments and found distinct differences in protein content based on the volume eluted from their SEC column. After characterization, the authors first treated immortalized hepatic stellate cells with established polarizing agents to either maintain a quiescent or activated state. In addition to the control treatments, the authors also treated with a potential therapeutic lipid-based formulation, S80, which has been previously implicated in decreasing rates of fibrogenesis within liver injury models. The authors then went through a systematic elimination protocol in order to isolate their candidate biomarker proteins, and chose SPARC as a proof-of concept molecule to assess their fluorescent NTA-methodologies.

Major overall concerns.

While the work presented is very interesting and the use of fluidic separation of EVs is in the direction the future of the field is heading, there are some gaps in the theory and conceptual design that should be improved upon. For example, what is the innovation of going through a highly sophisticated EV characterization, phenotype assessments, proteomics, and FL-NTA to test a protein that hasn't been assessed as being associated with EVs? I would recommend testing either one of the four TGF- β specific or any of the 74 R/PALM-specific proteins that may be associated with EVs in order for the authors to make their claims stronger. Also, one of the proteins from the inverse treatment group should be assessed with the FL-NTA. This would solidify that the proteomics method is robust and that EVs can definitively be used as biomarkers. While the basis of the studies is highly systematic and well-organized, some additional experiments are suggested below in order to strengthen the conclusions of the paper. Should these questions and concerns be addressed, the paper will be a strong candidate for publication in this journal.

Questions:

Figure 1.

- Authors have noticed a larger peak within their EAF4 samples after treatment with retinal and palmitic acid. This protein quantity is also higher in the SEC BCA quantification. Were these cells also cultured in serum-free medium, or is this an artifact of FBS?
- Were the treatment molecules well separated from their EVs under observation? If they weren't, is this flow field fractionation method sensitive enough to detect free TGF- β /Retinol-PA?

Figure 2.

- Because this method is not able to determine the impact of protein contaminants, a control should be included. Some examples could be by depleting EVs via size exclusion, filtration, dialysis, or something to show that the impact of the non-EV cell secretions are impacting oil red staining.
- "On the other hand, lipid droplets could still be found in cells treated using CCMb-S80 only, even though these cells were never in direct contact with the liposomal formulation"
 - o This point would be made stronger had the S80 formulation been fluorescently labeled. Washing cells of a hydrophobic material often needs a stronger agitation than PBS, since hydrophobic materials can insert into the cell membrane.
- "The newly stored lipid material must have been recycled through EVs released by the originally treated LX-2."
 - o No data were provided to support the above-mentioned claim.
- The images in figure 2 show that HSCs can self-assemble after perpetuation with TGF- β .
 - o There is a component in the conditioned media in the ROL/PA group that is inhibiting quiescence of the HSCs treated with Retinol/PALM, as well as the S80. This unknown factor or factors appear to be synergistic with TGF- β in forming self-assembled networks. This difference should be addressed in the main text in the results section under the appropriate figure.
- Quantification of genes should be included to show the perpetuation/activation of the HSCs in addition to the oil red assay showing quiescence. Is cell self-assembly in this model capable of being quantified? This could be a measure of HSEC activation. The investigators should repeat the experiment and treat the cells with compounds that decrease HSEC quiescence and increase HSEC activation in conjunction with S80 & R/PALM (HSEC Quiescence), and TGF β + Activating compound (HSEC Activation). This would further verify that the conditioned medium from TGF β treated cells is indeed causing HSEC activation.
- Minor: TGF- β concentrations in figure 2 are in wt/vol, whereas other treatments are in molar. It would be helpful to convert this to keep units consistent.

Figure 3.

- Panel C should be added to supplement or converted to a smaller heatmap for visualization

purposes.

- Please list out if not all the 78/4 common proteins, than just the 4 in the TGF-Beta group in panel F.

Figure 4.

- It might help for visualization to sort the pathway analysis with respect to the number of gene hits in each panel. For the Pathway graph, only 16 genes are represented out of the 44, are the other proteins unclassified? If they all only have one gene hit a pie-graph might be a better way of displaying this.

Figure 5.

- Please indicate which venn diagrams are control vs that of the treated groups.

Figure 6.

- Because the authors are arguing for using fluorescent NTA, they should include the NTA videos of the capture, instrument settings, as well as single dye controls to subtract out the dye/micelle populations with the PKH set of experiments.
- Panel F has some information that conflicts with previous observations seen in figure 2.
 - o Why is the number of SPARC+ particles so high for the HEPES group? Is this protein a stress response protein?
 - o Is the number of ROL/PA SPARC+ particles statistically different than that of the S80? Because the level of cell quiescence in figure 2 is comparable between the two groups, one would think that a secreted protein would be similar, but in Fig 6 panel F, these groups are very different. What is the explanation for this?
 - o If SPARC is to be used as a marker of cell phenotype, one would expect it to be statistically different between different cell phenotypes. It is recommended that the authors plot the CD81+ groups adjacent to each other, and the SPARC groups should be directly compared. The SPARC levels for different cell phenotypes would be expected to be significantly different from each other.
 - o Statistics should be included for this panel.

Reviewer #3 (Remarks to the Author):

In this study the authors examine extracellular vesicles (EVs) isolated from different treatment conditions of the hepatic stellate cell line LX-2 cells. Different EV characteristics were examined including a proteomic analysis. The SPARC protein was identified in EVs from LX-2 cells and this protein was used in a follow up proof of concept experiment using fluorescence nanoparticle tracking analysis. This is an exploratory study examining EVs isolated from LX-2 cells. The experiments are well-conducted and I have some comments that would strengthen the manuscript.

Rephrase in abstract "a solid set of methods".

I would expand upon the background on EVs in the introduction. For example, include information that EVs are released from cells and contain cargo etc. If you need to cut words I would cut some of the paragraph after it (this is redundant).

This is slightly confusing "Protein content associated with EVs was only detectable after 8 and 9 mL and was comparable in all groups." Do you mean 8 and 9 ml of conditioned media or after elution in SEC?

Figure 1a I would remove on the y-axis the label for negative particles. I would also include more labels on the x-axis. In addition, make the lines thicker and remove the SD. It is difficult to see the lines as is.

Figure 1b please make the lines thicker, they are tough to see.

In this line "After UC, up to 80% of total particles could be pelleted (Fig. S2). After SEC, EVs were successfully separated from protein aggregates co-purified during UC as determined by

bicinchoninic acid (BCA) assay (Fig. 1f-h)“ The Figures should be 1e-g. In these figures is the x-axis label the fraction # and/or elution from SEC? Typically (with IZON for example) SEC fractionates are 0.5 ml volumes so I would state in methods a/or legends that the elution fractions are 1 ml.

Proof of concept images and data should be shown that distinguishes that the different treatments elicit responses in the cells.

Fig. S2 what is “old UC protocol” versus “new UC protocol”? What is Mio stand for?

Table S1 are there significant differences between these different variables?

Figure S8, what is difference between EV pellet and SEC-fraction? Is EV-pellet from UC?

Figure 3b. Why is there white in the heat map? This is not shown in the color key.

Figure 3c, I am not sure if “persistent” is the correct term here, overlapping proteins?

Figure 3e. It is not clear here what 0 and 1 mean in the heat map? It is confusing the way it is described in the figure legend.

Figure 4. Programs that use gene ontology typically run from a large set of proteins. Therefore, it is not surprising that inputting 44 proteins would not yield any significant GO terms. It is more appropriate to use this analysis on the whole list of proteins identified. It is fine to keep Fig. 4, but be cautionary with interpretation. The manuscript would be strengthened (as mentioned in the discussion) by including network and pathway analysis for the protein lists.

“since it was one of the 4 proteins consistently found in TGF samples, but not in ROL/PA and S80 (Fig. 4f).” This should be 3f. Also, if this is the rationale I would point out all the 4 proteins that were found in TGF samples in 3f.

Figure 6f do you know if these proteins are co-labeled on the same EVs?

Methods please include information on the rotor and ultracentrifuge used.

For treatment of LX-2 cells with EVs, what was the dose or amount of EVs used to treat cells?

This sentence is missing something “EV-containing pellets originating from CCMa and CCMb were re-suspended after UC in fresh, serum-free DMEM, and they were then used to treat LX-2 cells seeded in 12-well plates the daybefore for 24 h, as well as to treat cells that were see.”

Staining of EVs with PKH67- was excess dye removed from the PKH-labeled EVs?

Reviewer #4 (Remarks to the Author):

Zivko et al present an interesting work in the development of a fluorescence-based NTA assay to identify secreted proteins that may be potentially used as non-invasive biomarkers of liver fibrogenesis. The authors characterized EVs isolated from cells undergoing the various treatments and found distinct differences in protein content based on the volume eluted from their SEC column. After characterization, the authors first treated immortalized hepatic stellate cells with established polarizing agents to either maintain a quiescent or activated state. In addition to the control treatments, the authors also treated with a potential therapeutic lipid-based formulation, S80, which has been previously implicated in decreasing rates of fibrogenesis within liver injury models. The authors then went through a systematic elimination protocol in order to isolate their candidate biomarker proteins, and chose SPARC as a proof-of concept molecule to assess their fluorescent NTA-methodologies.

Major overall concerns.

While the work presented is very interesting and the use of fluidic separation of EVs is in the direction the future of the field is heading, there are some gaps in the theory and conceptual design that should be improved upon. For example, what is the innovation of going through a highly sophisticated EV characterization, phenotype assessments, proteomics, and FL-NTA to test a protein that hasn't been assessed as being associated with EVs? I would recommend testing either one of the four TGF- β specific or any of the 74 R/PALM-specific proteins that may be associated with EVs in order for the authors to make their claims stronger. Also, one of the proteins from the inverse treatment group should be assessed with the FL-NTA. This would solidify that the proteomics method is robust and that EVs can definitively be used as biomarkers. While the basis of the studies is highly systematic and well-organized, some additional experiments are suggested below in order to strengthen the conclusions of the paper. Should these questions and concerns be addressed, the paper will be a strong candidate for publication in this journal.

Questions:

Figure 1.

- Authors have noticed a larger peak within their EAF4 samples after treatment with retinal and palmitic acid. This protein quantity is also higher in the SEC BCA quantification. Were these cells also cultured in serum-free medium, or is this an artifact of FBS?
- Were the treatment molecules well separated from their EVs under observation? If they weren't, is this flow field fractionation method sensitive enough to detect free TGF- β /Retinol-PA?

Figure 2.

- Because this method is not able to determine the impact of protein contaminants, a control should be included. Some examples could be by depleting EVs via size exclusion, filtration, dialysis, or something to show that the impact of the non-EV cell secretions are impacting oil red staining.
- "On the other hand, lipid droplets could still be found in cells treated using CCMb-S80 only, even though these cells were never in direct contact with the liposomal formulation"
 - o This point would be made stronger had the S80 formulation been fluorescently labeled. Washing cells of a hydrophobic material often needs a stronger agitation than PBS, since hydrophobic materials can insert into the cell membrane.
- "The newly stored lipid material must have been recycled through EVs released by the originally treated LX-2."
 - o No data were provided to support the above-mentioned claim.
- The images in figure 2 show that HSCs can self-assemble after perpetuation with TGF- β .
 - o There is a component in the conditioned media in the ROL/PA group that is inhibiting quiescence of the HSCs treated with Retinol/PALM, as well as the S80. This unknown factor or factors appear to be synergistic with TGF- β in forming self-assembled networks. This difference should be addressed in the main text in the results section under the appropriate figure.
- Quantification of genes should be included to show the perpetuation/activation of the HSCs in addition to the oil red assay showing quiescence. Is cell self-assembly in this model capable of being quantified? This could be a measure of HSEC activation. The investigators should repeat the experiment and treat the cells with compounds that decrease HSEC quiescence and increase HSEC activation in conjunction with S80 & R/PALM (HSEC Quiescence), and TGF β + Activating compound (HSEC Activation). This would further verify that the conditioned medium from TGF β treated cells is indeed causing HSEC activation.
- Minor: TGF- β concentrations in figure 2 are in wt/vol, whereas other treatments are in molar. It would be helpful to convert this to keep units consistent.

Figure 3.

- Panel C should be added to supplement or converted to a smaller heatmap for visualization purposes.
- Please list out if not all the 78/4 common proteins, than just the 4 in the TGF- β group in

panel F.

Figure 4.

- It might help for visualization to sort the pathway analysis with respect to the number of gene hits in each panel. For the Pathway graph, only 16 genes are represented out of the 44, are the other proteins unclassified? If they all only have one gene hit a pie-graph might be a better way of displaying this.

Figure 5.

- Please indicate which venn diagrams are control vs that of the treated groups.

Figure 6.

- Because the authors are arguing for using fluorescent NTA, they should include the NTA videos of the capture, instrument settings, as well as single dye controls to subtract out the dye/micelle populations with the PKH set of experiments.
- Panel F has some information that conflicts with previous observations seen in figure 2.
 - o Why is the number of SPARC+ particles so high for the HEPES group? Is this protein a stress response protein?
 - o Is the number of ROL/PA SPARC+ particles statistically different than that of the S80? Because the level of cell quiescence in figure 2 is comparable between the two groups, one would think that a secreted protein would be similar, but in Fig 6 panel F, these groups are very different. What is the explanation for this?
 - o If SPARC is to be used as a marker of cell phenotype, one would expect it to be statistically different between different cell phenotypes. It is recommended that the authors plot the CD81+ groups adjacent to each other, and the SPARC groups should be directly compared. The SPARC levels for different cell phenotypes would be expected to be significantly different from each other.
 - o Statistics should be included for this panel.

Point-by-point response to reviewers:

Reviewer #1 did not provide a full review but emailed some minor comments:

Minor comments to the authors

1) In the M&M section (EV isolation and purification) despite the authors nicely described the protocol, final details regarding volume of EV-pellet resuspension or SEC pooled fractions are missing. These data are necessary to understand the amount of particles used for SEM or cryo-TEM as well as extremely useful for the replication of the results by other groups. They may also serve as a starting reference for other related-projects. This data could be included in the Scheme figure 1.

We appreciate the reviewers' input. The volumes were added to Scheme 1 as suggested.

2) In the M&M section (Treatment of fresh LX-2 with EVs isolated from differentially treated LX-2), data regarding particle dose/volume dose are missing. These data are important for future replication of the results and reference points for other researchers. Did the authors try different EV particle concentration? Please include this information and clarify if dose-dependent effects were observed.

For this particular set of experiments, the whole pellet was used to avoid adding a sterilization step that would have otherwise been necessary. To do so, the most likely approach would have been the addition of a sterile filtration step, which would have inevitably lost us some particles, and would have required further validation. Having used the whole pellet, we did not have leftover sample for particle count, we can only infer from all other previous and subsequent experiments that the particle amounts were the same, i.e., around 10^{10} particles (see Table 1), in all instances given the consistency we were able to establish in our protocols.

We amended the method description to stress that the whole pellet was directly used (p. 25).

"The whole EV-containing pellets originating from CCMA and CCMB were directly re-suspended after UC in fresh, serum-free DMEM, and they were then used to treat LX-2 cells seeded in 12-well plates the day before for 24 h."

3) In the M&M section (Staining of EVs with PKH67), authors do not give details regarding potential removal of dye (washing steps) or any other considerations, especially important considering the controversial opinion about PKH67 results in the field.

Given our low EV yields in general, our problem was always the opposite, i.e., being below detection limits rather than having extra signal. PKH67 experiments were meant to show we could detect

fluorescently labelled EVs at all, before giving up on the optimization of labelling protocols with surface markers.

We did measure controls with just the dye in PBS with the same settings (see Methods), resulting in black NTA videos (one such video is attached for the reviewer, we can add it to the SI upon request, 2022_czivko_r01_nta02.avi).

Additionally, looking at the PKH67-labelled EVs data, two things show that we are measuring the same particles as before adding the dye, and not new particles coming from dye precipitates aggregating:

1) the size distribution profiles are more or less the same (see Fig. S16a, p.43), without new peaks of varying sizes.

2) The particle count is constant in scatter mode, while increasing in fluorescent mode upon increasing PKH67 concentration. We added a new plot in the SI reflecting this latter point (Fig. S16b, p. 43).

4) In the Results section, “EV isolation, purification and characterisation”, Fig. S2 shows a “change on UC protocol” on panel b. However, there is no mention about these changes in the manuscript. Could the authors clarify these experiments? New UC protocol may be the described one on M&M section, but, what was the “worse” one? It may be quite useful for other researchers in the field.

We apologize for the confusion: the graph was not meant for publication here. We do however truly appreciate the comment, and are including those details as they could indeed be useful to other researchers (p. 29).

“Fig. S2 | a,b, EV yield at different harvesting time points (a) and upon changing the UC protocol (b). For the old protocol, CCMs were centrifuged at 300 x g for 5 min at 4 °C, the supernatant was moved into a new tube and centrifuged again (10'000 x g, 20 min, 4 °C). The pelleted cell debris was discarded and the supernatant was ultracentrifuged (100'000 x g, 1 h 30 min, 4 °C). The three centrifugation steps were updated for the new UC protocols as described in the methods: 300 x g for 3 min, 9'000 x g for 30 min, 120'000 x g for 2 h 30 min. c, Comparison between the pelleted particles' yield and the particles' yield in the supernatant after UC. Mean ± SD, n = 2-3, software: NTA v3.2.”

5) In the Results section “Treatment of fresh LX-2 with EVs isolated from differentially treated LX-2”, it would be nice to refer to Fig. S3 directly on the text as well as on the Fig.2 caption. In addition, the authors claimed on the text “striking results from TGF treatment”, however, when Figure 2 is checked, TGF results do not seem so striking and if so, the results may be related to ROL/PA and not TGF. Please clarify.

References to Fig. S3 are added in the main text (p.5) and in the Fig.2 caption (p.6) as suggested.

“ORO staining was thus performed to reveal the presence of lipid droplets upon different treatments. Confirming our previous results, considerably more lipid droplets were identified in cells treated with S80 compared to any of the other treatments (Figs. 2a,b, *for additional details see Fig. S3*).”

“Fig. 1 | Treatment of fresh LX-2 with EV-pellets from previously treated cells. a, Representative images of ORO staining in fluorescence (seen as red spots; nuclei stained with blue DAPI) of differently treated cells after thresholding (see Fig. S3). b, Quantitative analysis of stained lipid droplets, whereby the fluorescent area (correlating to a quiescent-like status) was normalized to cell count (mean \pm SD, $n = 3$).”

As to the second part of the comment: the original text was remarking on visual similarities between direct treatment with S80, and treatment with EV-containing pellets originating from S80-treated cells, as well as similarities between direct treatment with TGF and treatment with EV-containing pellets originating from TGF-treated cells. For those instances, we found the trends observable by fluorescence microscopy to be “strikingly similar”.

We now removed the word “strikingly” from the main text to make it less confusing to the readers (p. 5)

“Cells that were treated with EV-containing pellets originating from the CCMB of TGF or S80-treated cells *had a response that was similar* to cells that were treated with TGF or S80 directly, albeit to a minor extent.”

6) In the same Results section, authors described a “structured network” but this is not easily appreciable on Figure 2 (or any other figure). May bright-field images help to this aim? Could the authors implement the Figure+panel references on the text? Since abbreviations are allowed, it may help the reader and not elongate too much the text.

Not expecting to see the cells rearranging themselves like that before treatment, and staining only with the goal of seeing nuclei and lipid droplets, the “network” described is only an observation emerging from cell nuclei spatial ordering. Brightfield images are sadly not helping in making this more appreciable, as it can be seen in Fig. S3.

We added Figure+panel references as suggested (p. 5).

“When looking at the effects of CCMB-TGF on naive cells, the microscopy images show that, remarkably, they arrange themselves along a structured network, much like TGF-treated cells (*Fig. 2a, panels TGF and CCM(b)-TGF*).”

7) Figure 2 results may be improved by breaking the Y-axis on panel Y to emphasise those results at lower fluorescence ratios. Connection to Supplementary Figure results maybe useful to understand.

The figure was modified as requested.

Reviewer #2 (Remarks to the Author):

Zivko et al present an interesting work in the development of a fluorescence-based NTA assay to identify secreted proteins that may be potentially used as non-invasive biomarkers of liver fibrogenesis. The authors characterized EVs isolated from cells undergoing the various treatments and found distinct differences in protein content based on the volume eluted from their SEC column. After characterization, the authors first treated immortalized hepatic stellate cells with established polarizing agents to either maintain a quiescent or activated state. In addition to the control treatments, the authors also treated with a potential therapeutic lipid-based formulation, S80, which has been previously implicated in decreasing rates of fibrogenesis within liver injury models. The authors then went through a systematic elimination protocol in order to isolate their candidate biomarker proteins, and chose SPARC as a proof-of concept molecule to assess their fluorescent NTA-methodologies.

Major overall concerns.

While the work presented is very interesting and the use of fluidic separation of EVs is in the direction the future of the field is heading, there are some gaps in the theory and conceptual design that should be improved upon. For example, what is the innovation of going through a highly sophisticated EV characterization, phenotype assessments, proteomics, and FL-NTA to test a protein that hasn't been assessed as being associated with EVs? I would recommend testing either one of the four TGF- β specific or any of the 74 R/PALM-specific proteins that may be associated with EVs in order for the authors to make their claims stronger. Also, one of the proteins from the inverse treatment group should be assessed with the FL-NTA. This would solidify that the proteomics method is robust and that EVs can definitively be used as biomarkers. While the basis of the studies is highly systematic and well-organized, some additional experiments are suggested below in order to strengthen the conclusions of the paper. Should these questions and concerns be addressed, the paper will be a strong candidate for publication in this journal.

We thank the reviewer for the positive evaluation of our work.

As far as the choice of the protein is concerned, SPARC was assessed as being associated with EVs exactly with our present work. The reviewer suggests testing one of the 4 proteins from the TGF group or one of the 74 from the ROL/PA group. SPARC is indeed one of the four proteins listed in the TGF group (Fig. 4f, p. 9). We added Table S2 to the SI (p. 46) to make this even clearer. Additionally, we did try our f-NTA method with 2 more proteins (GPC1 and IKKB) from the ROL/PA+S80 group, as mentioned in the discussion (p. 16):

“We have tried selecting proteins from the quiescent-like status list (Fig. 5f) which could be detected non-destructively, namely GPC1 and IKKB, but we were unable to detect effective immunolabeling with either one of them by f-NTA.”

However, it did not work with the current methods: the labelling might need protein specific optimization steps, or different incubation conditions based on the antibodies, and we did not investigate this aspect any further in the submitted study. It is also possible that the two proteins were not on the surface of the EVs, so permeabilization steps would have been necessary. We expanded upon these considerations in the discussion text to make it clearer for the reader that this was all already done (p. 16):

“It is possible that the labelling might need protein specific optimization steps, or that proteins were not on the surface of the EVs, requiring perhaps a permeabilization step in the protocol.”

Questions:

Figure 1.

- **Authors have noticed a larger peak within their EAF4 samples after treatment with retinal and palmitic acid. This protein quantity is also higher in the SEC BCA quantification. Were these cells also cultured in serum-free medium, or is this an artifact of FBS?**

Conditioned cell culture medium (CCM), from which EVs were harvested in every experiment, was always serum free, no FBS was ever used.

- **Were the treatment molecules well separated from their EVs under observation? If they weren't, is this flow field fractionation method sensitive enough to detect free TGF-Beta/Retinol-PA?**

They were not detectable. We had to substantially increase our EV-yield to even detect our EVs, which is one of the reasons why we needed a minimum of two T175 flasks per treatment condition for our experiments.

Figure 2.

- **Because this method is not able to determine the impact of protein contaminants, a control should be included. Some examples could be by depleting EVs via size exclusion, filtration, dialysis, or something to show that the impact of the non-EV cell secretions are impacting oil red staining.**

For this particular set of experiments, the whole pellet was used to avoid adding a sterilization step that would have otherwise been necessary (SEC purification was performed for everything else in the article).

The most likely approach would have been the addition of a sterile filtration step, which would have inevitably lost us some particles, and would have required further validation.

We never claimed that the observed effect was exclusively due to EVs and not to other non-EV components. This particular set of experiments was meant to see the effect of cells on neighbouring cells in general, and those findings prompted us to look into the role of EVs specifically later on (hence the proteomics on purified samples). Since our initial goal was to look into the possibility of using EVs as accessible biomarkers, our focus was not on what else might be used as such. We proved we can reliably isolate and purify EVs, a full validation for other components of the secretome was beyond the scope of the present study.

- **“On the other hand, lipid droplets could still be found in cells treated using CCMB-S80 only, even though these cells were never in direct contact with the liposomal formulation”**
 - o **This point would be made stronger had the S80 formulation been fluorescently labeled. Washing cells of a hydrophobic material often needs a stronger agitation than PBS, since hydrophobic materials can insert into the cell membrane.**

We appreciate the reviewer's input. There are three reasons why we did not label S80:

- 1) S80 is a complex mixture of many lipids extracted from plants, as we stated in the introduction. We would have had to validate homogenous labelling across batches of extracts, and establishing which components were labelled with which efficiency.
- 2) Any labelling molecule could have an additional effect on the cells in addition to the S80-effect we were trying to study.
- 3) The pure phospholipid DOPC control was added precisely to account for non-specific results due to hydrophobicity for example.

- **“The newly stored lipid material must have been recycled through EVs released by the originally treated LX-2.”**
 - o **No data were provided to support the above-mentioned claim.**

We apologise if our statement generated confusion. The sentence is our hypothesis based on the finding of stained lipid droplets in cells treated using CCMB-S80. We re-formulated to make it clearer in the text (p. 5):

*“The newly stored lipids **could be material that was recycled** through EVs released by the originally treated LX-2.”*

- **The images in figure 2 show that HSCs can self-assemble after perpetuation with TGF-Beta.**
 - o **There is a component in the conditioned media in the ROL/PA group that is inhibiting quiescence of the HSCs treated with Retinol/PALM, as well as the S80. This unknown factor or factors appear to be synergistic with TGF-Beta in forming self-assembled networks. This**

difference should be addressed in the main text in the results section under the appropriate figure.

LX-2 cells are in a semi-activated state at purchase (we refer to the Introduction and to ref. 28 in the manuscript). In 2010 the group of Prof. Scott Friedman at Mount Sinai School of Medicine (same group who immortalized human HSC generating the cell line LX-2) reported that treatments of retinol and palmitic acid promote downregulation of HSC activation via ADRP/PLIN2 upregulation (Lee et al. J Cell Physiol. (2010) 223: 648–657; doi: 10.1002/jcp.22063). We followed this protocol to de-activate transdifferentiated LX-2 cells and, as the unpublished data included in the next reply show, we could prove not only the upregulation of ADRP/PLIN2 upon ROL/PA treatment, but also that S80 treatment induces the same type of upregulation associated with the LX-2 lipid droplets. We thus do not believe that the CCM in the ROL/PA group could inhibit the quiescence of the HSCs in the ROL/PA group, but quite the opposite. We apologies if we misunderstood the comment.

• **Quantification of genes should be included to show the perpetuation/activation of the HSCs in addition to the oil red assay showing quiescence. Is cell self-assembly in this model capable of being quantified? This could be a measure of HSEC activation.**

Perpetuation and activation in the LX-2 in vitro model are well established before in the literature we cite in the manuscript (ref. 28 in the manuscript), including our group's previous work (ref. 29).

Our research groups also have preliminary mRNA expression results (see below, unpublished data for reviewers' eyes only, please keep confidential) to that effect, which are part of a different project carried out by a scientist not co-author of the present manuscript. We hope the previous scientific article(s) and these additional preliminary data sufficiently address the reviewer's concerns.

About the figure above: Alpha-1 type I collagen (**COL1A1**) is overexpressed in fibrotic HSCs, Perilipin-2 (**PLIN2**) codes for the adipose differentiation-related protein (ADRP) which is associated to lipid droplets and overexpressed in quiescent HSCs, and secreted protein acidic and cysteine rich (**SPARC**) which translates into SPARC that was shown in the present work to be underexpressed in quiescent HSCs

The investigators should repeat the experiment and treat the cells with compounds that decrease HSEC quiescence and increase HSEC activation in conjunction with S80 & R/PALM (HSEC Quiescence), and TGFB + Activating compound (HSEC Activation). This would further verify that the conditioned medium from TGFB treated cells is indeed causing HSEC activation.

We thank the reviewer for the valuable suggestion. The choice of compounds to deactivate the transdifferentiation (ROL/PA) and to perpetuate the fibrogenesis (TGF) of LX-2 were chosen because well established, as our characterization and the above shown mRNA data confirm. No antifibrotic treatment is to date known and using specific chemicals would activate only specific pathways, which in turn would provide only a limited snapshot of a complex process such as fibrogenesis and perpetuation of fibrosis, often orchestrated by neighboring cells via a mechanotransduction-mediated microenvironmental stiffness (Kostallari et al, Am J Physiol Gastrointest Liver Physiol. (2022) 322(2):G234-G246. doi: 10.1152/ajpgi.00254.2021; Sakai et al, Biol. Pharm. Bull. (2021) 44:416–421, doi:10.1248/bpb.b20-00815).

• Minor: TGF-Beta concentrations in figure 2 are in wt/vol, whereas other treatments are in molar. It would be helpful to convert this to keep units consistent.

We added the molarity in the Methods (p. 23). Since protein concentrations are usually expressed in wt/v, we left that as well for clarity, hopefully the reviewer will not mind.

“Cells were then washed with phosphate buffered saline (PBS) and treated for 24 h with different solutions prepared in serum free cell culture media (DMEM): either ROL/PA (10/300 μ M), TGF (10 ng/mL, 227.27 pM), or liposomal formulations (5 mM lipid concentration) of S80 or DOPC.”

Figure 3.

• Panel C should be added to supplement or converted to a smaller heatmap for visualization purposes.

Does the reviewer mean Fig. 3b, i.e., Panel B, instead of C? (Fig. 3c is a table). If so, we now added it as Fig. S19 for better visualization.

• Please list out if not all the 78/4 common proteins, than just the 4 in the TGF-Beta group in panel F.

We thank the reviewer for the suggestion. Proteins are listed in the newly added Table S2.

Figure 4. (now Fig. S10!)

- It might help for visualization to sort the pathway analysis with respect to the number of gene hits in each panel. For the Pathway graph, only 16 genes are represented out of the 44, are the other proteins unclassified? If they all only have one gene hit a pie-graph might be a better way of displaying this.

It was mostly sparse one gene hits, which is unsurprising given the small set, as another reviewer pointed out. We moved the figure to SI to reflect its lesser bearing on our study.

This means that Figure 4 from the previous manuscript version is now Fig. S10, and Figures 5 and 6 from the previous manuscript version are now 4 and 5 respectively. We will point this out in the rest of the comments as a reminder.

Figure 5. (now Fig. 4!)

- Please indicate which venn diagrams are control vs that of the treated groups.

Out of the 6 tested conditions (DMEM, ROL/PA, TGF, HEPES, S80, DOPC), 5 are different types of control (see Scheme 1) for the S80 treatment under investigation.

The Venn diagrams in the figure are not comparing treatments vs controls, they are meant to compare SEC-purified samples (which had only 1 EV-containing peak) to AF4-purified samples (with two EV-containing peaks, p1 and p2 respectively) for each of the 6 conditions.

Figure 6. (now Fig. 5!)

- Because the authors are arguing for using fluorescent NTA, they should include the NTA videos of the capture, instrument settings, as well as single dye controls to subtract out the dye/micelle populations with the PKH set of experiments.

We appreciate the input. A representative NTA video is now attached (2022_czivko_r01_nta01.avi) and instrument settings are detailed in the methods (p. 24).

“(NTA, ZetaView 8.05.05 SP2 equipped with a 488 nm laser, zetapotential and temperature control units, Particle Metrix). Measurements were performed at 25 °C, a camera sensitivity of 80, and 100 ms⁻¹ shutter value. Particles were traced for at least 15 consecutive frames, videos were taken at 11 positions. Samples had to have ≥ 200 traced particles.”

As to the concerns regarding dye aggregates: given our low EV yields in general, our problem was always the opposite, i.e., being below detection limits rather than having extra signal. PKH67 experiments were meant to show we could detect fluorescently labelled EVs at all, before giving up on the optimization of labelling protocols with surface markers.

We did measure controls with just the dye in PBS with the same settings (see Methods), resulting in black NTA videos (one such video is attached for the reviewer, we can add it to the SI upon request, 2022_czivko_r01_nta02.avi).

Additionally, looking at the PKH67-labelled EVs data, two things show that we are measuring the same particles as before adding the dye, and not new particles coming from dye precipitates aggregating:

- 1) the size distribution profiles are more or less the same (see Fig. S16a, p.43), without new peaks of varying sizes.
- 2) The particle count is constant in scatter mode, while increasing in fluorescent mode upon increasing PKH67 concentration. We added a new plot in the SI reflecting this latter point (Fig. S16b, p. 43).

• **Panel F has some information that conflicts with previous observations seen in figure 2.**

o **Why is the number of SPARC+ particles** so high for the HEPES group? Is this protein a stress response protein?

Yes, it is our major contention that SPARC on EVs can be used to measure stress response in our in vitro model. It was indeed one of the 4 candidate proteins emerging in the TGF group from (see Fig. 4f and Table S2). 10% v/v of HEPES buffer is an important dilution of cell culture medium, especially for a 24 h treatment. The stress it caused on cells, while not measurably apparent from Fig. 2, can now be measured through the number of SPARC+ EVs.

o **Is the number of ROL/PA SPARC+ particles statistically different than that of the S80? Because the level of cell quiescence in figure 2 is comparable between the two groups, one would think that a secreted protein would be similar, but in Fig 6 panel F, these groups are very different. What is the explanation for this?**

We were hesitant to perform statistical analysis with a sample size of n=3, but we added it as requested to show that if performed, it does show statistical significance here.

As the reviewer points out, the “level of cell quiescence” from ORO staining of cytoplasmic lipid droplets looked similar, but the proteomic profiles were very different (and the SPARC+ EVs trends are also similar but not the same). The explanation for this is multi-layered:

- 1) Analysis of microscopy images is semi-quantitative: we made an effort to quantify what we see in the images, but it's never going to be a precise measure.
- 2) Cytoplasmic lipid droplets and composition of EVs are different types of phenotypical response.
- 2) ROL/PA is not used as treatment for liver fibrosis, essential phospholipids are, even though their mechanism of action is poorly understood (we refer to Introduction). It was our working hypothesis that proteomic profiling of EVs could gain us novel insights into it, and indeed it did.
- 3) S80 does have an incredible effect, we found a way to show it by EV-analysis, which can be implemented as a novel and convenient screening tool. It is also telling something new about the so far unexplained mechanism of action of the S80 complex mixture, i.e., it has an effect resulting in a dramatic lowering of EV-associated SPARC presence (we refer to Results and Discussion).

o If SPARC is to be used as a marker of cell phenotype, one would expect it to be statistically different between different cell phenotypes. It is recommended that the authors plot the CD81+ groups adjacent to each other, and the SPARC groups should be directly compared. The SPARC levels for different cell phenotypes would be expected to be significantly different from each other.

o Statistics should be included for this panel.

We agree with the reviewer that the CD81+ groups should be adjacent to each other, and that the SPARC group should be directly compared, as we originally showed in the submitted Figure 6f (now 5f). In case the reviewer was referring to a figure splitting, we report below the two plots next to each other for reviewing purposes only:

We would rather keep the two plots together in the same Figure, so the readers can immediately see how CD81 levels are similar in all samples, regardless of treatment condition, while SPARC presence on EVs changes based on treatment.

We included statistical analysis it in the graph as suggested and we added a “Statistics and reproducibility” paragraph in the Methods.

Reviewer #3 (Remarks to the Author):

In this study the authors examine extracellular vesicles (EVs) isolated from different treatment conditions of the hepatic stellate cell line LX-2 cells. Different EV characteristics were examined including a proteomic analysis. The SPARC protein was identified in EVs from LX-2 cells and this protein was used in a follow up proof of concept experiment using fluorescence nanoparticle tracking analysis. This is an exploratory study examining EVs isolated from LX-2 cells. The experiments are well-conducted and I have some comments that would strengthen the manuscript.

Rephrase in abstract “a solid set of methods”.

I would expand upon the background on EVs in the introduction. For example, include information that EVs are released from cells and contain cargo etc. If you need to cut words I would cut some of the paragraph after it (this is redundant).

The text was changed as suggested (p. 2).

“Extracellular vesicles (EVs) are membranous nanosized vesicles mediating inter-cellular communication, secreted by virtually all cells²⁰⁻²³. They are increasingly being investigated for their potential as diagnostic tools given the rich differences that can arise in their biochemical composition as well as in their cargo²⁴⁻²⁶.”

This is slightly confusing “Protein content associated with EVs was only detectable after 8 and 9 mL and was comparable in all groups.” Do you mean 8 and 9 ml of conditioned media or after elution in SEC?

After elution in SEC, we clarified in the text (p.3).

“Protein content associated with EVs was only detectable after 8 and 9 mL of elution upon SEC, and was comparable in all groups.”

Figure 1a I would remove on the y-axis the label for negative particles. I would also include more labels on the x-axis. In addition, make the lines thicker and remove the SD. It is difficult to see the lines as is.

Figure 1b please make the lines thicker, they are tough to see.

The figure was improved as suggested.

In this line “After UC, up to 80% of total particles could be pelleted (Fig. S2). After SEC, EVs were successfully separated from protein aggregates co-purified during UC as determined by bicinchoninic acid (BCA) assay (Fig. 1f-h)” The Figures should be 1e-g. In these figures is the x-axis label the fraction # and/or elution from SEC? Typically (with IZON for example) SEC

fractionates are 0.5 ml volumes so I would state in methods a/or legends that the elution fractions are 1 ml.

Figure axes shows “eluted volume [mL]”. The volume of the eluted fractions was 1 mL: we also added this information to Scheme 1 in the Methods for clarity.

We corrected Fig. 1f-h to Fig. 1e-g (p.3).

“After SEC, EVs were successfully separated from protein aggregates co-purified during UC as determined by bicinchoninic acid (BCA) assay (Fig. 1e-g).”

Proof of concept images and data should be shown that distinguishes that the different treatments elicit responses in the cells.

The baseline treatments (DMEM, ROL/PA and TGF) are established in the literature (ref. 28 in the manuscript), including our group’s previous work (ref. 29). In the present study we expended this to include the effect that differently treated LX-2 cells have on their neighbouring cells (Fig.2), as well as proteomic profiling of their EVs (Fig. 3-5).

Fig. S2 what is “old UC protocol” versus “new UC protocol”? What is Mio stand for?

We apologize for the confusion: the graph was not meant for publication here. We do however truly appreciate the comment, and are including those details as they could indeed be useful to other researchers (p. 29).

“Fig. S2 | a,b, EV yield at different harvesting time points (a) and upon changing the UC protocol (b). For the old protocol, CCMs were centrifuged at 300 x g for 5 min at 4 °C, the supernatant was moved into a new tube and centrifuged again (10’000 x g, 20 min, 4 °C). The pelleted cell debris was discarded and the supernatant was ultracentrifuged (100’000 x g, 1 h 30 min, 4 °C). The three centrifugation steps were updated for the new UC protocols as described in the methods: 300 x g for 3 min, 9’000 x g for 30 min, 120’000 x g for 2 h 30 min. c, Comparison between the pelleted particles’ yield and the particles’ yield in the supernatant after UC. Mean ± SD, n = 2-3, software: NTA v3.2.”

Mio stands for millions, we changed it to “x 10⁶” for clarity.

Table S1 are there significant differences between these different variables?

No, there were no significant difference when looking at EV average yield, size and Zeta Potential.

Figure S8, what is difference between EV pellet and SEC-fraction? Is EV-pellet from UC?

Yes, see Scheme 1 in the methods: EV-pellets are samples re-suspended after UC, which were either used as such or which were further purified by SEC.

Figure 3b. Why is there white in the heat map? This is not shown in the color key.

It's the 0 value, i.e., no hit detected. We did perform imputation (see Fig. S11) to fill in the gaps, but we thought it's more honest to show the panel as it is in the main manuscript, especially with a sample size of n=3.

Figure 3c, I am not sure if “persistent” is the correct term here, overlapping proteins?

By persistent we meant that it was persistently detected across biologically independent samples, i.e., it was found in all 3 of them, and not just 1 or 2. We do hope that this explanation clarified the use of this adjective.

Figure 3e. It is not clear here what 0 and 1 mean in the heat map? It is confusing the way it is described in the figure legend.

We thank the reviewer for the comment. We expand upon the explanation of the normalization of expression scores used to generate the heat map in Fig. 3e (p. 9).

“For ease of comparison, a simple, normalized recovery score was developed by adding the LFQ values of every protein for each treatment condition and normalizing it to the sum of all of them, so that the panel could be visually inspected as a heat map.”

Figure 4. Programs that use gene ontology typically run from a large set of proteins. Therefore, it is not surprising that inputting 44 proteins would not yield any significant GO terms. It is more appropriate to use this analysis on the whole list of proteins identified. It is fine to keep Fig. 4, but be cautionary with interpretation. The manuscript would be strengthened (as mentioned in the discussion) by including network and pathway analysis for the protein lists.

We agree with the reviewer, this is not surprising considering the small input. We decided to move the panel to the SI (now Fig. S10) to reflect its limited bearing on this study without further network and pathway analyses, which would be out of the scope of the current manuscript.

“since it was one of the 4 proteins consistently found in TGF samples, but not in ROL/PA and S80 (Fig. 4f).” This should be 3f. Also, if this is the rationale I would point out all the 4 proteins that were found in TGF samples in 3f.

We thank the reviewer for drawing our attention to this oversight. We corrected “Fig. 4f” to “Fig. 3f”. Proteins are listed in the newly added Table S2 as requested.

Figure 6f do you know if these proteins are co-labeled on the same EVs?

Sadly not. Co-localization was not technologically possible on NTA at the time these experiments were performed due to the position of the filters in the instruments. Proteomic analysis only shows that they are found on the same samples, but does not tell us what is found on each of the single EVs.

Methods please include information on the rotor and ultracentrifuge used.

The information was included as requested (p. 23).

“The pelleted cell debris was discarded and the supernatant was ultracentrifuged (120'000 x g, 2 h 30 min, 4 °C, Beckman Coulter, Optima XPN Ultracentrifuge, Type 70 Ti rotor or SW 32 Ti rotor).”

For treatment of LX-2 cells with EVs, what was the dose or amount of EVs used to treat cells?

For this particular set of experiments, the whole pellet was used to avoid adding a sterilization step that would have otherwise been necessary. To do so, the most likely approach would have been the addition of a sterile filtration step, which would have inevitably lost us some particles, and would have required further validation. Having used the whole pellet, we did not have leftover sample for particle count, we can only infer from all other previous and subsequent experiments that the particle amounts were the same, i.e., around 10^{10} particles (see Table 1), in all instances given the consistency we were able to establish in our protocols.

We amended the method description to stress that the whole pellet was directly used (p. 25):

“The whole EV-containing pellets originating from CCMA and CCMB were directly re-suspended after UC in fresh, serum-free DMEM, and they were then used to treat LX-2 cells seeded in 12-well plates the day before for 24 h.”

This sentence is missing something “EV-containing pellets originating from CCMA and CCMB were re-suspended after UC in fresh, serum-free DMEM, and they were then used to treat LX-2 cells seeded in 12-well plates the day before for 24 h, as well as to treat cells that were see.”

The sentence was corrected (we refer to the previous comment's response).

Staining of EVs with PKH67- was excess dye removed from the PKH-labeled EVs?

It was not. Given our low EV yields in general, our problem was always the opposite, i.e., being below detection limits rather than having extra signal. PKH67 experiments were meant to show we could detect fluorescently labelled EVs at all, before giving up on the optimization of labelling protocols with surface markers.

We did measure controls with just the dye in PBS with the same settings (see Methods), resulting in black NTA videos (one such video is attached for the reviewer, we can add it to the SI upon request, 2022_czivko_r01_nta02.avi).

Additionally, looking at the PKH67-labelled EVs data, two things show that we are measuring the same particles as before adding the dye, and not new particles coming from dye precipitates aggregating:

- 1) the size distribution profiles are more or less the same (see Fig. S16a, p.43), without new peaks of varying sizes.
- 2) The particle count is constant in scatter mode, while increasing in fluorescent mode upon increasing PKH67 concentration. We added a new plot in the SI reflecting this latter point (Fig. S16b, p. 43).

Reviewer #4 (Remarks to the Author):

We believe that the comments of reviewer #4 are the same as of reviewer #2, so we refer to our point-to-point reply above.

Zivko et al present an interesting work in the development of a fluorescence-based NTA assay to identify secreted proteins that may be potentially used as non-invasive biomarkers of liver fibrogenesis. The authors characterized EVs isolated from cells undergoing the various treatments and found distinct differences in protein content based on the volume eluted from their SEC column. After characterization, the authors first treated immortalized hepatic stellate cells with established polarizing agents to either maintain a quiescent or activated state. In addition to the control treatments, the authors also treated with a potential therapeutic lipid-based formulation, S80, which has been previously implicated in decreasing rates of fibrogenesis within liver injury models. The authors then went through a systematic elimination protocol in order to isolate their candidate biomarker proteins, and chose SPARC as a proof-of concept molecule to assess their fluorescent NTA-methodologies.

Major overall concerns.

While the work presented is very interesting and the use of fluidic separation of EVs is in the direction the future of the field is heading, there are some gaps in the theory and conceptual design that should be improved upon. For example, what is the innovation of going through a highly sophisticated EV characterization, phenotype assessments, proteomics, and FL-NTA to test a protein that hasn't been assessed as being associated with EVs? I would recommend testing either one of the four TGF-B specific or any of the 74 R/PALM-specific proteins that may be associated with EVs in order for the authors to make their claims stronger. Also, one of the proteins from the inverse treatment group should be assessed with the FL-NTA. This would solidify that the proteomics method is robust and that EVs can definitively be used as biomarkers. While the basis of the studies is highly systematic and well-organized, some additional experiments are suggested below in order to strengthen the conclusions of the paper. Should these questions and concerns be addressed, the paper will be a strong candidate for publication in this journal.

Questions:

Figure 1.

- Authors have noticed a larger peak within their EAF4 samples after treatment with retinal and palmitic acid. This protein quantity is also higher in the SEC BCA quantification. Were these cells also cultured in serum-free medium, or is this an artifact of FBS?

- Were the treatment molecules well separated from their EVs under observation? If they weren't, is this flow field fractionation method sensitive enough to detect free TGF-Beta/Retinol-PA?

Figure 2.

- Because this method is not able to determine the impact of protein contaminants, a control should be included. Some examples could be by depleting EVs via size exclusion, filtration, dialysis, or something to show that the impact of the non-EV cell secretions are impacting oil red staining.
- “On the other hand, lipid droplets could still be found in cells treated using CCMB-S80 only, even though these cells were never in direct contact with the liposomal formulation”
 - o This point would be made stronger had the S80 formulation been fluorescently labeled. Washing cells of a hydrophobic material often needs a stronger agitation than PBS, since hydrophobic materials can insert into the cell membrane.
- “The newly stored lipid material must have been recycled through EVs released by the originally treated LX-2.”
 - o No data were provided to support the above-mentioned claim.
- The images in figure 2 show that HSCs can self-assemble after perpetuation with TGF-Beta.
 - o There is a component in the conditioned media in the ROL/PA group that is inhibiting quiescence of the HSCs treated with Retinol/PALM, as well as the S80. This unknown factor or factors appear to be synergistic with TGF-Beta in forming self-assembled networks. This difference should be addressed in the main text in the results section under the appropriate figure.
- Quantification of genes should be included to show the perpetuation/activation of the HSCs in addition to the oil red assay showing quiescence. Is cell self-assembly in this model capable of being quantified? This could be a measure of HSEC activation. The investigators should repeat the experiment and treat the cells with compounds that decrease HSEC quiescence and increase HSEC activation in conjunction with S80 & R/PALM (HSEC Quiescence), and TGFB + Activating compound (HSEC Activation). This would further verify that the conditioned medium from TGFB treated cells is indeed causing HSEC activation.
- Minor: TGF-Beta concentrations in figure 2 are in wt/vol, whereas other treatments are in molar. It would be helpful to convert this to keep units consistent.

Figure 3.

- Panel C should be added to supplement or converted to a smaller heatmap for visualization purposes.
- Please list out if not all the 78/4 common proteins, than just the 4 in the TGF-Beta group in panel F.

Figure 4.

- It might help for visualization to sort the pathway analysis with respect to the number of gene hits in each panel. For the Pathway graph, only 16 genes are represented out of the 44, are the other proteins unclassified? If they all only have one gene hit a pie-graph might be a better way of displaying this.

Figure 5.

- Please indicate which venn diagrams are control vs that of the treated groups.

Figure 6.

- Because the authors are arguing for using fluorescent NTA, they should include the NTA videos of the capture, instrument settings, as well as single dye controls to subtract out the dye/micelle populations with the PKH set of experiments.
- Panel F has some information that conflicts with previous observations seen in figure 2.
 - o Why is the number of SPARC+ particles so high for the HEPES group? Is this protein a stress response protein?
 - o Is the number of ROL/PA SPARC+ particles statistically different than that of the S80? Because the level of cell quiescence in figure 2 is comparable between the two groups, one would think that a secreted protein would be similar, but in Fig 6 panel F, these groups are very different. What is the explanation for this?
 - o If SPARC is to be used as a marker of cell phenotype, one would expect it to be statistically different between different cell phenotypes. It is recommended that the authors plot the CD81+ groups adjacent to each other, and the SPARC groups should be directly compared. The SPARC levels for different cell phenotypes would be expected to be significantly different from each other.
 - o Statistics should be included for this panel.

REVIEWERS' COMMENTS:

Reviewer #2 (Remarks to the Author):

The authors have satisfactorily addressed all of our comments. I recommend this paper for publication.

Reviewer #3 (Remarks to the Author):

The authors have addressed my concerns and the manuscript is acceptable for publication.